COMMUNICATIONS

# Constraining climate sensitivity and continental versus seafloor weathering using an inverse geological carbon cycle model

Joshua Krissansen-Totton[1] & David C. Catling[1]

The relative influences of tectonics, continental weathering and seafloor weathering in controlling the geological carbon cycle are unknown. Here we develop a new carbon cycle model that explicitly captures the kinetics of seafloor weathering to investigate carbon fluxes and the evolution of atmospheric $CO_2$ and ocean pH since 100 Myr ago. We compare model outputs to proxy data, and rigorously constrain model parameters using Bayesian inverse methods. Assuming our forward model is an accurate representation of the carbon cycle, to fit proxies the temperature dependence of continental weathering must be weaker than commonly assumed. We find that 15–31 °C (1σ) surface warming is required to double the continental weathering flux, versus 3–10 °C in previous work. In addition, continental weatherability has increased 1.7–3.3 times since 100 Myr ago, demanding explanation by uplift and sea-level changes. The average Earth system climate sensitivity is $5.6^{+1.3}_{-1.2}$ K (1σ) per $CO_2$ doubling, which is notably higher than fast-feedback estimates. These conclusions are robust to assumptions about outgassing, modern fluxes and seafloor weathering kinetics.

[1] Department of Earth and Space Sciences/Astrobiology Program, University of Washington, Seattle, Washington 98195-1310, USA. Correspondence and requests for materials should be addressed to J.K.-T. (email: joshkt@uw.edu).

Determining what controls the geological carbon cycle is crucial for understanding climate stability, planetary habitability and the long-term consequences of anthropogenic carbon emissions. On long timescales, carbon inputs into the atmosphere–ocean system must balance outputs otherwise atmospheric $CO_2$ would be depleted leading to a runaway icehouse, or $CO_2$ would accumulate and become excessively abundant within 10–100 Myr ago (ref. 1). Thus, a negative feedback must stabilize the carbon cycle and global climate on geological timescales.

The carbonate–silicate weathering cycle described by Walker et al.[2] is widely believed to provide a negative feedback[3]. In this picture, climatic warming from abundant atmospheric $CO_2$ enhances silicate weathering and delivers more carbon to the ocean where it precipitates to form carbonate rocks. Conversely, climatic cooling due to low $CO_2$ reduces silicate weathering and dampens $CO_2$ drawdown. The temperature dependence of silicate weathering effectively provides a natural thermostat to stabilize climate[2].

Although the carbonate–silicate thermostat is now part of textbook Earth science[4], the link between global climate, $CO_2$ and silicate weathering rates is unclear. Gaillardet et al.[5] found no overall correlation between weathering rates and temperature on a global scale. Some regional field studies also fail to find a relationship[6]. Furthermore, strongly temperature-dependent continental weathering is argued to contradict the conventional interpretation of the Phanerozoic strontium isotope record[7]. Weatherability changes due to uplift, lithology and biology have all been proposed as alternative drivers of the carbon cycle[8–11]. Indeed, Walker[12] himself changed his opinion to suspecting greater sensitivity of weathering rate to the aforementioned factors than to $CO_2$. Some models of the Cenozoic carbon cycle also suggest sizeable weatherability changes for reproducing observed $pCO_2$ and carbon isotope histories[13].

However, weatherability changes alone do not provide a clear negative feedback to balance the carbon cycle. An alternative or complimentary negative feedback to continental silicate weathering is seafloor weathering[7,14–17]. Seafloor weathering occurs because seawater circulates through upper oceanic crust in low-temperature, off-axis hydrothermal systems. Reactions with basalt release calcium ions, which precipitates calcium carbonate within veins and pores of the oceanic crust[18,19]. If the rate of basalt dissolution and precipitation is linked—directly or indirectly—to the carbon content of the atmosphere–ocean system, then seafloor weathering could provide an important negative feedback.

Here it is useful to think in terms of carbonate alkalinity, $[HCO_3^-] + 2[CO_3^{2-}]$, which—neglecting the small contribution from weak acid anions and water dissociation products—must balance the sum of conservative cations minus conservative anions, $([Na^+] + 2[Mg^{2+}] + 2[Ca^{2+}] + [K^+] + \ldots - [Cl^-] - 2[SO_4^{2-}] - [Br^-] - \ldots)$. Thus, dissolution of basalt, which releases alkaline earth and alkali metal cations, generates carbonate alkalinity to neutralize the positive ions in solution. In other words, carbon speciation adjusts in response to the addition of conservative cations to ensure charge balance.

Caldeira[20] used simple geochemical models to show that the pH dependence of seafloor weathering is too weak to be an important carbon cycle feedback, concluding that continental weathering dominates. However, recent studies challenge this conclusion: Mesozoic-aged oceanic crust has substantially higher carbonate content compared to Cenozoic-aged cores[18], and this elevated carbonate content is due to enhanced Mesozoic carbonate precipitation and not carbonate accumulation later[21]. In addition, these seafloor carbonates can only be explained by alkalinity released from basalt dissolution; alkalinity released from continental weathering is insufficient[16]. Temperature-dependent seafloor weathering could explain both the observed change in seafloor carbonate abundance since the Mesozoic and the strontium isotope record[17]. These studies justify re-examining the importance of seafloor weathering in the global carbon cycle.

Previous attempts to incorporate seafloor weathering into carbon cycle models have produced conflicting results, and temperature dependencies are often omitted[7,20]. Brady and Gíslason[14] conducted experiments to find the $pCO_2$ dependence and temperature dependence of seafloor weathering rates, which they summarized as:

$$r_{\text{dissolution}} \propto (R_{CO_2})^\mu = \left(\frac{pCO_2}{pCO_2^{\text{mod}}}\right)^\mu \qquad (1)$$

Here $R_{CO_2}$ is the partial pressure of atmospheric $CO_2$ ($pCO_2$) relative to preindustrial modern ($pCO_2^{\text{mod}} = 280$ p.p.m.), $\mu = 0.23$ is the best-fit $CO_2$ dependence and $\mu = 0.32$ is the best-fit temperature dependence. This approximation, with varying values for $\mu$, has been adopted in several subsequent models, which suggest that seafloor weathering was an important negative feedback during at least some of Earth's history[10,15,22].

Subsuming the pH dependence, temperature dependence and ocean chemistry dependence of seafloor weathering into a $CO_2$ dependence (such as equation (1)) is potentially problematic because it does not explicitly capture dissolution kinetics, and so the value for $\mu$ that combines the temperature, pH and ocean chemistry weathering dependencies must be either guessed or fitted (for example, ref. 15). Models such as GEOCARB and its various incarnations[3,23] similarly do not incorporate seafloor weathering accurately because they lack ocean chemistry. Some models attempt to capture dissolution kinetics. However, these models have highly simplified ocean chemistry[24], assume a constant temperature of dissolution[25] or do not reproduce the observed increase[17] in seafloor basalt dissolution[26].

Here we develop a new open source model for testing competing hypotheses about the global carbon cycle. Our model explicitly calculates ocean chemistry and includes pH-dependent and temperature-dependent kinetics based on laboratory and empirical studies. This is an improvement over previous models that subsumed pH dependence and temperature dependence into an overall indirect $CO_2$ dependence with an arbitrary functional relationship (equation (1)), or only considered pH dependence in isolation[20,25]. In addition, we use a new parameterization linking deep-ocean temperatures to surface temperatures, and thus seafloor weathering to climate. The Cenozoic (66–0 Myr ago) and Mesozoic (252–66 Myr ago) are particularly useful eras to model because of their relatively abundant proxy records of ocean composition and geochemistry. Thus, to validate our model, we apply it the last 100 Myr ago and compare model outputs with these proxies. We also use a Bayesian inversion to constrain model parameters quantitatively, including the temperature sensitivity of continental weathering, climate sensitivity, and continental weatherability changes. This approach makes minimal assumptions about carbon cycle processes. For instance, it is agnostic about whether Cretaceous outgassing was very high or comparable to present levels, and it allows for large changes in weatherability of both silicates and carbonates. Consequently, the novel probability distributions we obtain for parameters of interest are robust to model assumptions.

## Results

**Definition of parameters**. We model the time evolution of the carbon cycle using two separate boxes representing the atmosphere–ocean and the pore-space in the seafloor (Fig. 1), loosely following Caldeira[20]. We track carbon and alkalinity

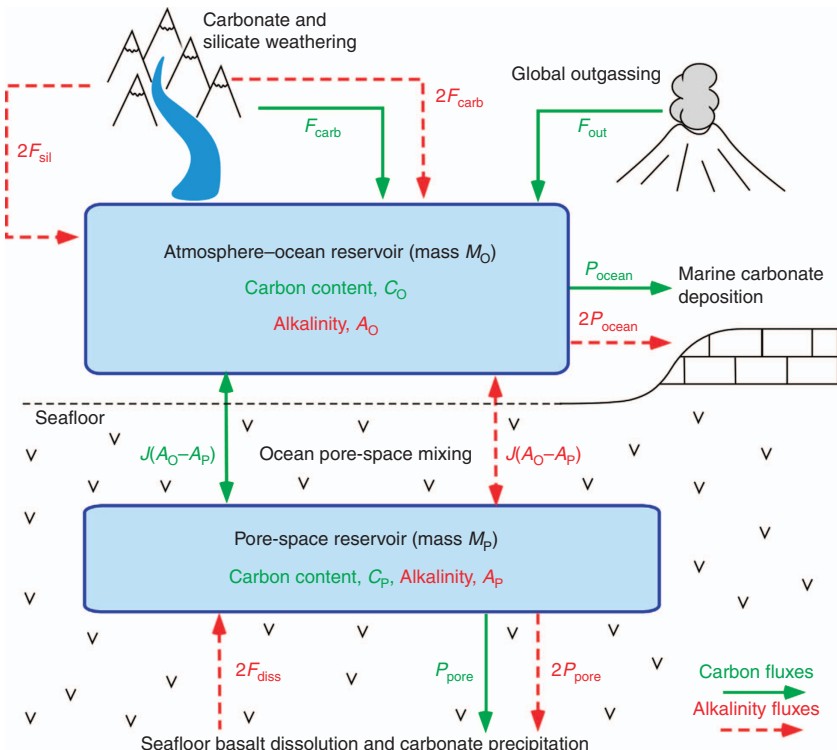

**Figure 1 | Schematic representation of the box model used in this study.** Carbon fluxes (Tmol C per year) are denoted by solid-green arrows, and alkalinity fluxes (Tmol eq per year) are denoted by red-dashed arrows. The fluxes into/out of the atmosphere–ocean are outgassing, $F_{out}$, silicate weathering, $F_{sil}$, carbonate weathering, $F_{carb}$, and marine carbonate precipitation, $P_{ocean}$. The fluxes into/out of the pore-space are basalt dissolution, $F_{diss}$, and pore-space carbonate precipitation, $P_{pore}$. Alkalinity fluxes are multiplied by two because the uptake or release of one mole of carbon as carbonate is balanced by a cation with a $2+$ charge (typically $Ca^{2+}$). A constant mixing flux, $J$ (kg per year), exchanges carbon and alkalinity between the atmosphere–ocean and pore-space.

fluxes into and between those boxes, and specify a constant mixing flux, $J$ (kg per year) that determines how rapidly the ocean circulates through the pore-space. We assume that the bulk ocean is in equilibrium with the atmosphere. Consequently, our model is only appropriate for timescales greater than the mixing time of the ocean, $\sim 1,000$ years.

For many free parameters, we assume a range of values rather than point estimates. Model outputs are presented as distributions over this range of parameter values, which ensures that our results are robust to uncertainties in parameters, as described below. The dynamical equations, flux parameterizations and initial conditions are described in the methods section. Here we summarize the key points necessary to understand the results.

Following Walker *et al.*[2], the continental silicate weathering flux is expressed as:

$$F_{sil} = \omega F_{sil}^{mod} \left( \frac{pCO_2}{pCO_2^{mod}} \right)^{\alpha} \exp(\Delta T_S / T_e) \qquad (2)$$

Here $\Delta T_S = T_S - T_S^{mod}$ is the difference in global mean surface temperatures relative to preindustrial modern, $T_S^{mod} = 285\,K$; $\omega$ is a dimensionless weatherability factor, $F_{sil}^{mod}$ is the modern silicate weathering flux (Tmol per year) and $\alpha$ is an empirical constant (see Methods). An e-folding temperature, $T_e$, defines the temperature dependence of weathering, which can be related to an effective activation energy, $E_a$, as follows:

$$\frac{1}{T_e} \approx \frac{E_a}{RT_S^2} \qquad (3)$$

Here $R$ is the universal gas constant.

In equation (2), we have combined both the direct kinetic temperature dependence and the temperature dependence of

runoff in a single exponent. This is justifiable because an exponential kinetic dependence and a linear runoff dependence can be accurately approximated by an overall exponential (Supplementary Fig. 1). Walker *et al.*[2] also combine both effects to obtain an overall temperature dependence of $T_e = 13.7\,K$, which corresponds to an effective activation energy of $\sim 50\,kJ\,mol^{-1}$. Similarly, field measurements of chemical weathering fluxes from field studies suggest an activation energy of $74 \pm 29\,kJ\,mol^{-1}$ ($T_e = 9.1^{+5.9}_{-2.6}\,K$ (ref. 27)). GEOCARB III effectively uses an overall e-folding temperature of $T_e \sim 9.2\,K$, combining a direct temperature dependence and runoff coefficient[3]. We initially assume $T_e = 5-15\,K$, but later consider a broader range because a weak temperature dependence turns out to be required.

The factor $\omega$ represents all so-called 'external' variables affecting weatherability and encompasses changes in land area due to sea-level variations, changes in lithology, relief, biology and palaeogeography. Rather than attempt to explicitly model these different influences (for example, ref. 23), we address the inverse problem: how much does $\omega$ have to change to fit proxy data? The simplest approach is to assume a linear change in weatherability since the mid-Cretaceous, as follows,

$$\omega = 1 + W(t/100\,Myr) \qquad (4)$$

and find $W$, the unknown change in weatherability. Here $t$ is millions of years ago (Myr ago). We initially assume $W = 0$ to examine the range of possible model outputs for no external weatherability changes. Later, we assume $W = -0.8$ to 0.2, corresponding to mid-Cretaceous weatherability that was between 80% less or 20% greater than modern.

We adopt a parameterized climate model, whereby changes in surface temperature, $\Delta T_S$, are logarithmically related to $pCO_2$, with corrections for solar luminosity evolution and palaeogeography (see Methods). The climate sensitivity parameter, $\Delta T_{2x}$, has units of Kelvin warming per $CO_2$ doubling. The Intergovernmental Panel on Climate Change (IPCC)[28] estimated equilibrium climate sensitivity within the range 1.5–4.5 K. We assume $\Delta T_{2x} = 1.5$–8.0 K, which encompasses a broad range.

**Approach and context.** Parameterizations for carbon cycle processes are uncertain, so we adopted two statistical techniques to extract robust conclusions. First, we ran the forward model for the range of parameters and initial values shown in Tables 1 and 2. Each range was sampled as a uniform distribution, and the forward model was run 10,000 times to build distributions for the time series model outputs such as $pCO_2$, pH and temperatures. The resulting distributions represent the full range of possible model outcomes given assumed parameterizations and parameter ranges. These distributions were then compared to proxy data, typically binned into 10 Myr ago intervals with large, conservative error bars (Supplementary Methods).

The second approach solved the inverse problem with Markov Chain Monte Carlo (MCMC) techniques using the *emcee* package[29] in python (Supplementary Methods). Here the parameter ranges and initial value ranges define uniform priors (Table 1). A likelihood function describes the goodness-of-fit with proxy data, and the forward model was run 10 million times to obtain posterior probability distributions for the model parameters. The inverse solution gauges the full extent to which proxy data constrain the operation of the carbon cycle. In particular, parameters such as climate sensitivity, weatherability changes and the temperature dependence of silicate weathering can be constrained.

**Forward modelling.** The forward model was run 10,000 times, repeatedly sampling the parameter ranges of Table 1. First,

weatherability was assumed constant ($W = 0$) and a range for the temperature dependence of silicate weathering typically assumed in the literature was adopted ($T_e = 5$–15 K). Figure 2 shows the modelled 90% confidence intervals plotted alongside proxy data. Clearly, the model with commonly assumed ranges for parameters does not fit the data. In particular, deep-ocean temperatures, surface temperatures and seafloor carbonate precipitation at 100 Myr ago are considerably lower than Cretaceous proxies (Fig. 2d,f). Fits with pH and ocean saturation state are also poor.

Next, the calculation was repeated using a weaker temperature dependence of silicate weathering, $T_e = 30$–40 K (Fig. 3). Here the 90% model uncertainty envelopes broadly encompass the proxy data, but the fit is marginal. The model $pCO_2$ output is consistent with $pCO_2$ proxies, and the temperature and seafloor precipitation distributions mostly overlap with proxy data error bars. Similarly, if we repeat the calculation a third time with the original temperature dependence of silicate weathering ($T_e = 5$–15 K), but include sizeable weatherability changes ($W = -0.6$ to $-0.4$, that is, weatherability at 100 Myr ago between 40 and 60% the modern value), then the model plausibly fits the data (Supplementary Fig. 2). Temperature, $pCO_2$ and precipitation proxies fall broadly within the model uncertainty envelope, although the fits with ocean pH and saturation state are still poor.

The best fit (Fig. 4) is obtained by assuming both weak temperature dependence of silicate weathering ($T_e = 30$–40 K) and a large weatherability change ($W = -0.6$ to $-0.4$). The median model output then fits temperature and precipitation proxies well, and $pCO_2$, ocean saturation state and pH all fall within proxy error bars.

From these results, we conclude that either the temperature dependence of silicate weathering is weaker than is commonly assumed, and/or that weatherability has approximately doubled since 100 Myr ago. When both are true, the model best fits the

**Table 1 | Parameter ranges and Bayesian Markov Chain Monte Carlo inversion results.**

| Variable | Prior (uniform) | Nominal model | | | Michaelis–Menten law | |
|---|---|---|---|---|---|---|
| | | Median posterior value | 68% Credible interval $(1 - \sigma)$ | 90% Credible interval | Median posterior value | 90% Credible interval |
| $CO_2$ dependence, $\alpha^*$ | 0.2–0.5 | 0.33 | 0.24–0.44 | 0.21–0.48 | 0.41 | 0.05–0.91 |
| e-folding temp. dep. of cont. weathering, $T_e$ (K) | 5–50 | 34 | 22–45 | 17–48 | 31 | 14–48 |
| Relative Cretaceous, weatherability, $1 + W$ | 0.2–1.2 | 0.42 | 0.30–0.58 | 0.24–0.71 | 0.6 | 0.31–0.96 |
| Climate sensitivity, $\Delta T_{2x}$ (K) | 1.5–8.0 | 5.6 | 4.4–6.9 | 3.7–7.5 | 5.6 | 3.8–7.6 |
| Relative Cretaceous outgassing, $1 + V$ | 1.2–2.5 | 1.58 | 1.34–1.88 | 1.25–2.1 | 1.56 | 1.24–2.1 |
| Carbonate weatherability modifier, $1 + C_{WF}$ | 0.1–2.5 | 0.36 | 0.18–0.63 | 0.13–0.83 | 0.39 | 0.13–0.88 |
| Modern outgassing, $F_{out}^{mod}$ (Tmol C per year) | 4–10 | 6.6 | 4.8–8.8 | 4.2–9.6 | 6.5 | 4.2–9.6 |
| Modern carb. weathering, $F_{carb}^{mod}$ (Tmol C per year) | 7–14 | 11 | 8.2–12.9 | 7.4–13.7 | 11 | 7.4–13.7 |
| Pore-space circulation time, $\tau$ (kyr) | 20–1,000 | 570 | 239–847 | 100–949 | 555 | 88–945 |
| Carbonate precip. coefficient, $n$ | 1.0–2.5 | 1.66 | 1.22–2.18 | 1.1–2.4 | 1.69 | 1.1–2.4 |
| Modern seafloor dissolution relative to precipitation, $x^\dagger$ | 0.5–1.5 | 1.02 | 0.68–1.34 | 0.56–1.45 | 1.01 | 0.56–1.45 |
| Surface-deep temp. gradient, $a_{grad}$ | 0.8–1.4 | 0.99 | 0.88–1.14 | 0.83–1.25 | 0.99 | 0.83–1.25 |
| pH dependence seafloor, $\gamma$ | 0–0.5 | 0.27 | 0.11–0.43 | 0.04–0.48 | 0.27 | 0.04–0.48 |
| Temp. dependence seafloor, $E_{bas}$ (kJ mol$^{-1}$) | 40–110 | 75 | 53–97 | 45–106 | 76 | 45–106 |
| Modern pelagic fraction, $f_{PEL}$ | 0.4–0.6 | 0.49 | 0.43–0.56 | 0.41–0.59 | 0.49 | 0.41–0.59 |
| Spreading rate dep., $\beta$ | 0.0–1.0 | 0.47 | 0.15–0.82 | 0.05–0.94 | 0.49 | 0.05–0.95 |
| Palaeogeography climate parameter, $\Delta P$ (K) | 0.0–5.0 | 2.6 | 0.88–4.2 | 0.28–4.7 | 2.5 | 0.27–4.7 |

Column 1 shows the variables we wish to constrain using proxy data. Column 2 gives the uniform prior for each variable. These intervals also constitute the ranges assumed in the forward model analysis, unless stated otherwise. Columns 3–5 describe the posterior probability distributions for each variable for the nominal model. Columns 6–7 describe the posterior probability distributions for the modified model where the $pCO_2$ dependence of continental weathering is parameterized using the Michaelis–Menten law.
*For the Michaelis–Menten law, the prior for $\alpha$ is 0–1.0 (Supplementary Note 6), and so the posterior distribution is different to that of the nominal model.
†This constant defines the initial seafloor dissolution flux relative to the carbonate precipitation flux in the pore-space (see Table 2 for further details and formal definition).

**Table 2 | Initial values or initial value ranges assumed in our model.**

| Variable | Initial value or initial range | References |
|---|---|---|
| Modern pore-space carbonate precipitation, $P_{pore}^{mod}$ (Tmol C per year) | 0.45* | [18] (see Supplementary Methods) |
| Modern seafloor dissolution, $F_{diss}^{mod}$ (Tmol C per year) | 0.225–0.675† | [16] |
| Modern outgassing, $F_{out}^{mod}$ (Tmol C per year) | 4–10 | [23,76] |
| Modern carb. Weathering, $F_{carb}^{mod}$ (Tmol C per year) | 7–14 | [77], their Table 2 |
| Preindustrial mean surface temperature, $T_S$ (K) | 285 | [2] |
| Modern ocean pH | 8.2 | [74] |
| Ocean Ca abundance (mMol kg$^{-1}$) | 10.03 | Polynomial fit to Tyrrell and Zeebe[75], see Supplementary Fig. 8 |
| Preindustrial atmospheric pCO$_2$ (p.p.m.) | 280 | — |
| Modern fraction pelagic precip. $f_{PEL}$ | 0.4–0.6 | [78] |

*Because we are adopting wide ranges for $F_{out}^{mod}$ and $F_{carb}^{mod}$, it is unnecessary to include a range for $P_{pore}^{mod}$ because its size relative to outgassing and weathering fluxes already encompasses a wide range (only the relative sizes of carbon cycle fluxes matter for predicting observable variables).
†Here we assume that $F_{diss}^{mod} = xP_{pore}^{mod}$, where $x = 0.5$–1.5. Coogan and Gillis[16] used a geochemical model of pore-space precipitation to show that at least 70% of pore-space precipitation is attributable to alkalinity release from basalt dissolution. Here we conservatively assume a lower limit of 50% instead. The upper limit is 150% to allow for the possibility that pore-space dissolution exceeds pore-space precipitation, and that the excess alkalinity is mixed into the ocean to form marine carbonates.

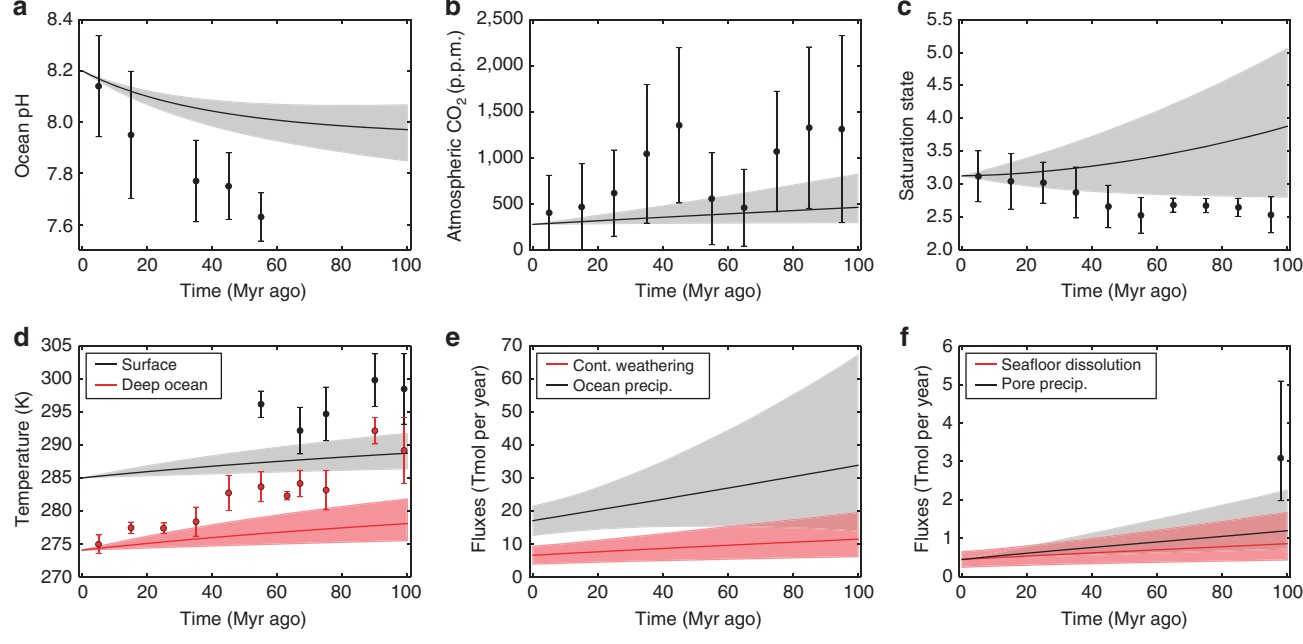

**Figure 2 | Carbon cycle model with poor fit to data assuming conventional temperature dependence of continental weathering and no weatherability change.** Selected model outputs and geochemical proxy data for a conventional temperature sensitivity range for continental weathering ($T_e = 5$–15 K), and no change in silicate weatherability over the last 100 Myr ago ($W = 0$). Grey- and red-shaded regions represent the model output 90% confidence obtained from 10,000 forward model runs using the parameter ranges described in Table 1. The grey- and red-solid lines are the median model outputs. Black and red dots represent binned geochemical proxy data, and error bars denote the range of binned proxy estimates (see main text for references and explanation). Panels denote (**a**) ocean pH, (**b**) atmospheric pCO$_2$, (**c**) ocean saturation state, (**d**) mean surface and deep ocean temperatures, (**e**) continental silicate weathering and ocean carbonate precipitation fluxes, and (**f**) seafloor dissolution and pore space carbonate precipitation fluxes. This case is a very poor fit to temperature and pH, and is a relatively poor fit to ocean saturation state and seafloor carbonate precipitation.

data. This result is robust to carbon cycle assumptions because the model distributions were calculated assuming the full parameter ranges in Table 1.

**Bayesian MCMC inversion.** The forward model results are qualitatively instructive, but Bayesian analysis allows more quantitative conclusions. MCMC techniques generated Fig. 5, which shows 95% credible intervals for the time evolution of carbon cycle variables. MCMC produces a much better fit than forward modelling because the algorithm converges to the maximum-likelihood region of parameter space. The distribution of model outcomes fits every

proxy we considered within the 95% interval except one ocean pH data point. Distributions are also plotted for the relative and absolute change in seafloor dissolution, continental silicate weathering and continental carbonate weathering.

Figure 6 shows posterior probability distributions for selected model parameters, and Table 1 shows the best estimates for all parameters with uncertainty. To test the sensitivity of our results to weathering parameterizations, results are reported for both the nominal model (power-law pCO$_2$ dependence of continental weathering) and a Michaelis–Menten pCO$_2$ dependence law (Supplementary Note 6). For the nominal model, the temperature

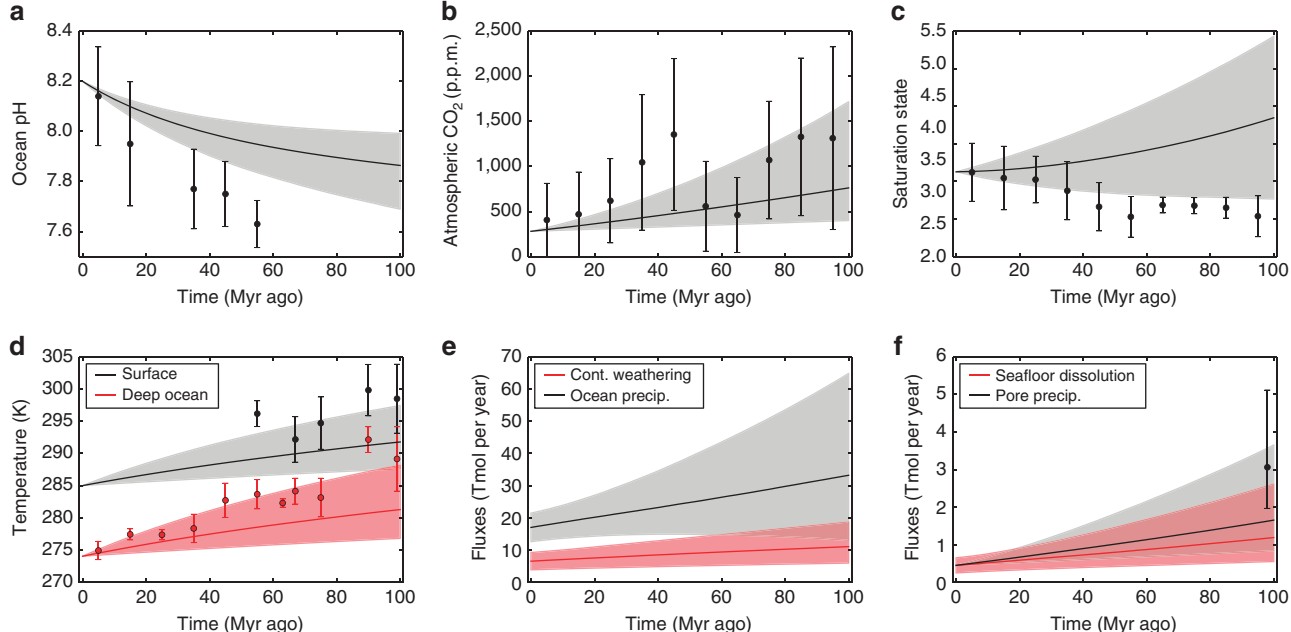

**Figure 3 | Carbon cycle model with moderate fit to data assuming weak temperature dependence of continental weathering and no weatherability change.** Selected model outputs and geochemical proxy data for a weak temperature dependence for continental weathering ($T_e = 30$–40 K) and no change in silicate weatherability over the last 100 Myr ago ($W = 0$). Grey- and red-shaded regions represent the model output 90% confidence obtained from 10,000 forward model runs using the parameter ranges described in Table 1. The grey- and red-solid lines are the median model outputs. Black and red dots represent binned geochemical proxy data, and error bars denote the range of binned proxy estimates (see main text for references and explanation). Panels denote (**a**) ocean pH, (**b**) atmospheric $pCO_2$, (**c**) ocean saturation state, (**d**) mean surface and deep ocean temperatures, (**e**) continental silicate weathering and ocean carbonate precipitation fluxes, and (**f**) seafloor dissolution and pore space carbonate precipitation fluxes. Here the model envelopes marginally encompass the proxy data. The upper end of the temperature and seafloor envelopes fit proxies, $pCO_2$ is an excellent fit, and the saturation state and ocean pH proxies are on the edge of the model envelope.

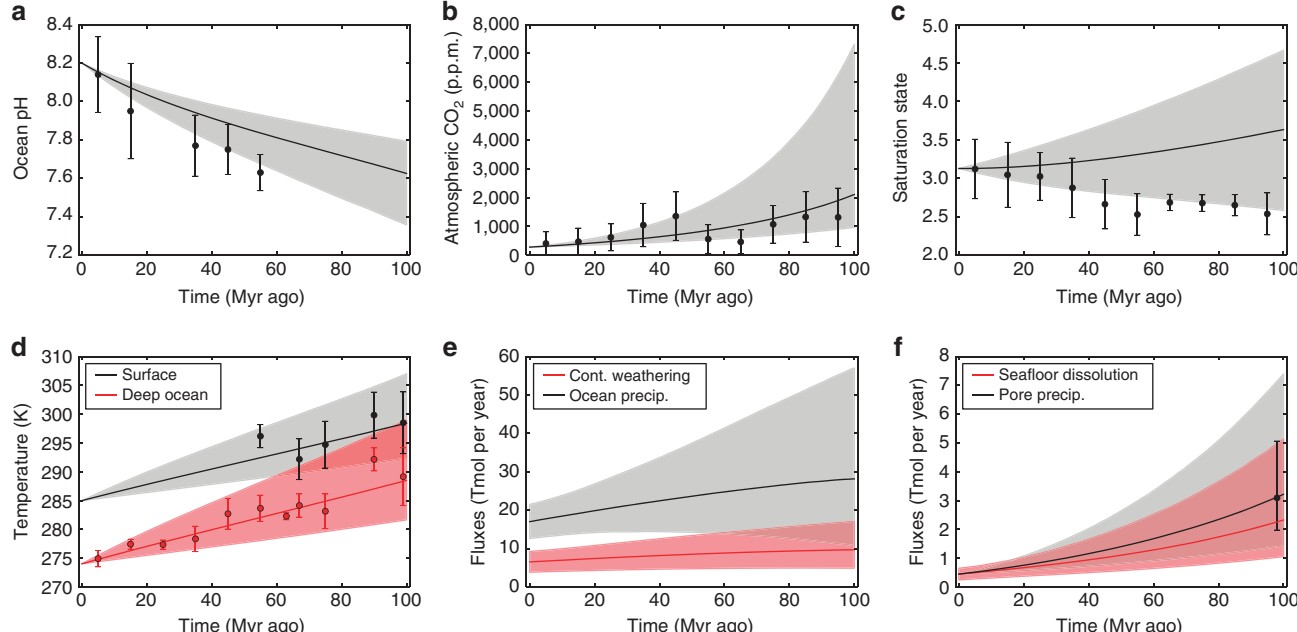

**Figure 4 | Carbon cycle model with excellent fit to data assuming weak temperature dependence of continental weathering and a weatherability doubling since 100 Myr ago.** Selected model outputs and geochemical proxy data for a weak temperature dependence for continental weathering ($T_e = 30$–40 K) and a 40–60% change in continental weatherability over the last 100 Myr ago ($W = -0.6$ to $-0.4$). Grey- and red-shaded regions represent the model output 90% confidence obtained from 10,000 forward model runs using the parameter ranges described in Table 1. The grey- and red-solid lines are the median model outputs. Black and red dots represent binned geochemical proxy data, and error bars denote the range of binned proxy estimates (see main text for references and explanation). Panels denote (**a**) ocean pH, (**b**) atmospheric $pCO_2$, (**c**) ocean saturation state, (**d**) mean surface and deep ocean temperatures, (**e**) continental silicate weathering and ocean carbonate precipitation fluxes, and (**f**) seafloor dissolution and pore space carbonate precipitation fluxes. Here the model envelopes are an excellent fit with proxy data. The median temperature and seafloor precipitation approximately coincide with geochemical proxies, and the saturation state, pH and $pCO_2$ envelopes all encompass their respective proxies.

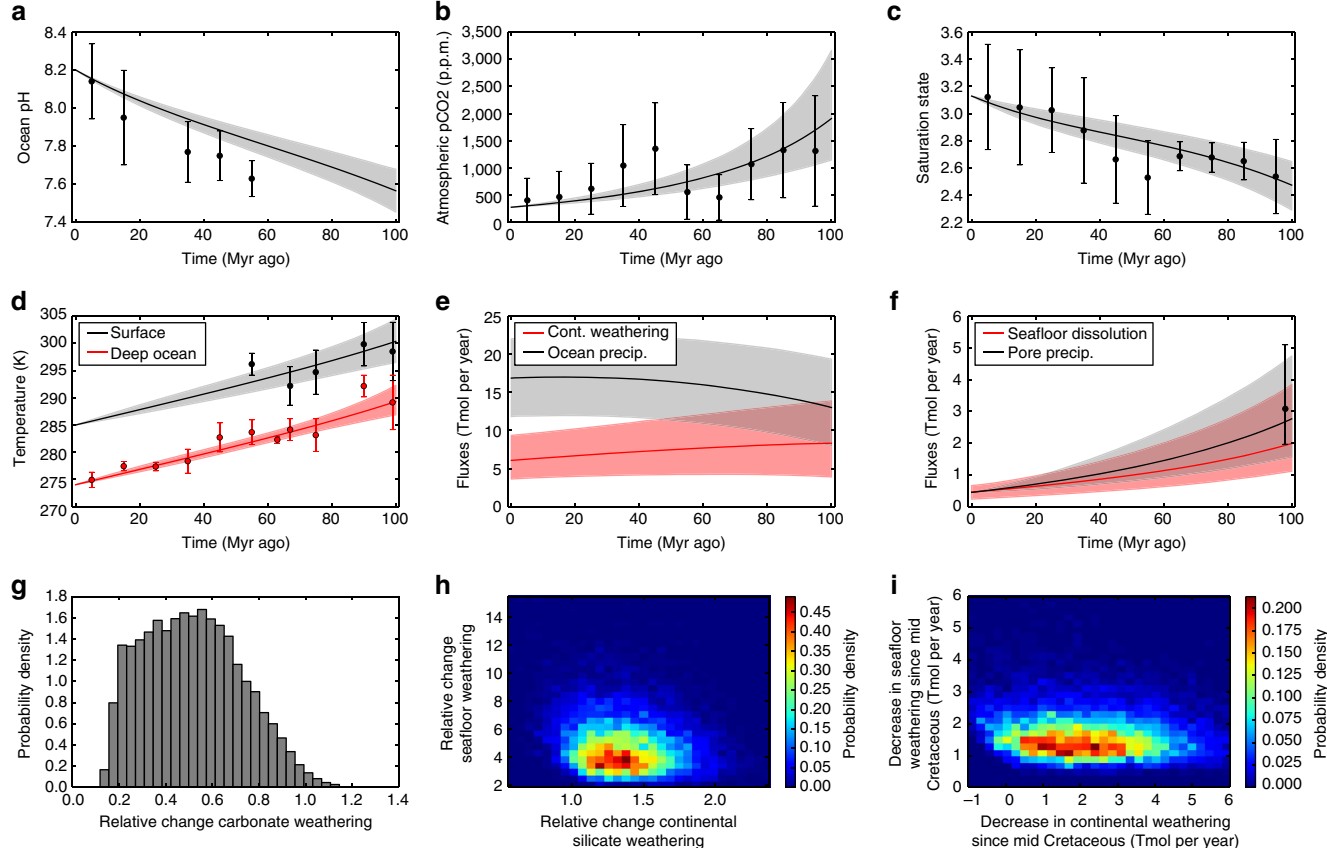

**Figure 5 | Carbon cycle model fitted to data with Bayesian inverse analysis.** Bayesian Markov Chain Monte Carlo (MCMC) results for the nominal model. (**a**–**f**) Grey- and red-shaded areas show 95% credible intervals for selected model outputs, and solid lines show median model outputs. Black and red circles are geochemical proxies, and error bars denote the range of binned proxy estimates (see main text). Note that the credible intervals encompass the Cretaceous proxies (within uncertainty) in almost all cases. (**g**–**i**) Probability distributions for the relative and absolute change in seafloor dissolution, continental silicate weathering and continental carbonate weathering. Relative changes refer to flux at 100 Myr ago relative to the modern flux.

dependence of silicate weathering is $T_e = 34^{+11}_{-11}$ K ($1\sigma$). This corresponds to a low effective activation energy of $20^{+10}_{-5}$ kJ mol$^{-1}$ ($1\sigma$). Note that this effective activation energy incorporates all hydrological cycle feedbacks, not just the direct kinetic effect of temperature. Moreover, there is a 98% probability that $T_e \geq 15$ K ($E_a \leq 45$ kJ mol$^{-1}$). This suggests that the widely used range[2,3] of $T_e = 5$–15 K is incorrect. Intuitively, continental weathering must be weakly dependent on temperature because otherwise temperatures and pCO$_2$ would be too low, and ocean pH and saturation state values would be too high compared to geochemical proxies (Fig. 2).

The Bayesian inversion also implies that silicate weatherability in the Cretaceous was $42^{+16}_{-12}$% ($1\sigma$) of modern weatherability for the nominal model (Fig. 6). In addition, we can say there is a 95% probability that Cretaceous weatherability was $\leq 71$% of modern weatherability. This increase in weatherability is required in part because without it, Cretaceous silicate weathering would be too high to allow for the large observed seafloor weathering sink of carbon. But even if seafloor weathering is assumed to be negligible and removed from the likelihood function, then Cretaceous weatherability is $46^{+16}_{-13}$% modern weatherability. This small change is surprising because including seafloor weathering and its Cretaceous constraint effectively imposes a decreasing carbon sink, thereby allowing for a larger weatherability increase since 100 Myr ago and a weaker temperature sensitivity of continental weathering. However, the seafloor weathering sink is small compared to continental weathering, and so its omission only subtly affects the inversion. Instead, it is mostly temperature,

pCO$_2$, pH and saturation state proxies that constrain the weatherability change.

The average equilibrium climate sensitivity over the last 100 Myr ago is constrained to $\Delta T_{2x} = 5.6^{+1.3}_{-1.2}$ K per CO$_2$ doubling ($1\sigma$) in the nominal model (Fig. 6). The 90% credible interval extends from 3.7 to 7.5 K, which is much higher than IPCC estimates. Low climate sensitivity is precluded by a lack of fit to pCO$_2$ and temperature proxies. The inverse analysis also suggests that Cretaceous outgassing was unlikely ($\sim$9% probability) to be greater than double modern outgassing to fit pCO$_2$ and temperature proxies.

In addition to the nominal model for continental weathering (equation (2)), we repeated the inverse analysis replacing the pCO$_2$ dependence of continental weathering with the Michaelis–Menten law:

$$F_{sil} = \omega F_{sil}^{mod}\left(\frac{2R_{CO_2}}{1 + R_{CO_2}}\right)^\alpha \exp(\Delta T_S/T_e) \qquad (5)$$

Using this modified parameterization, the results described above are largely unchanged (column 2, Fig. 6). The biggest difference between the two models is that the weatherability change required since 100 Myr ago is likely more modest under the Michaelis–Menten law than the nominal model, with a median Cretaceous value of 0.6 rather than 0.42.

Figure 5h shows the joint probability distribution for the relative change in continental silicate and seafloor weathering. We see the relative change in seafloor basalt dissolution at 100 Myr

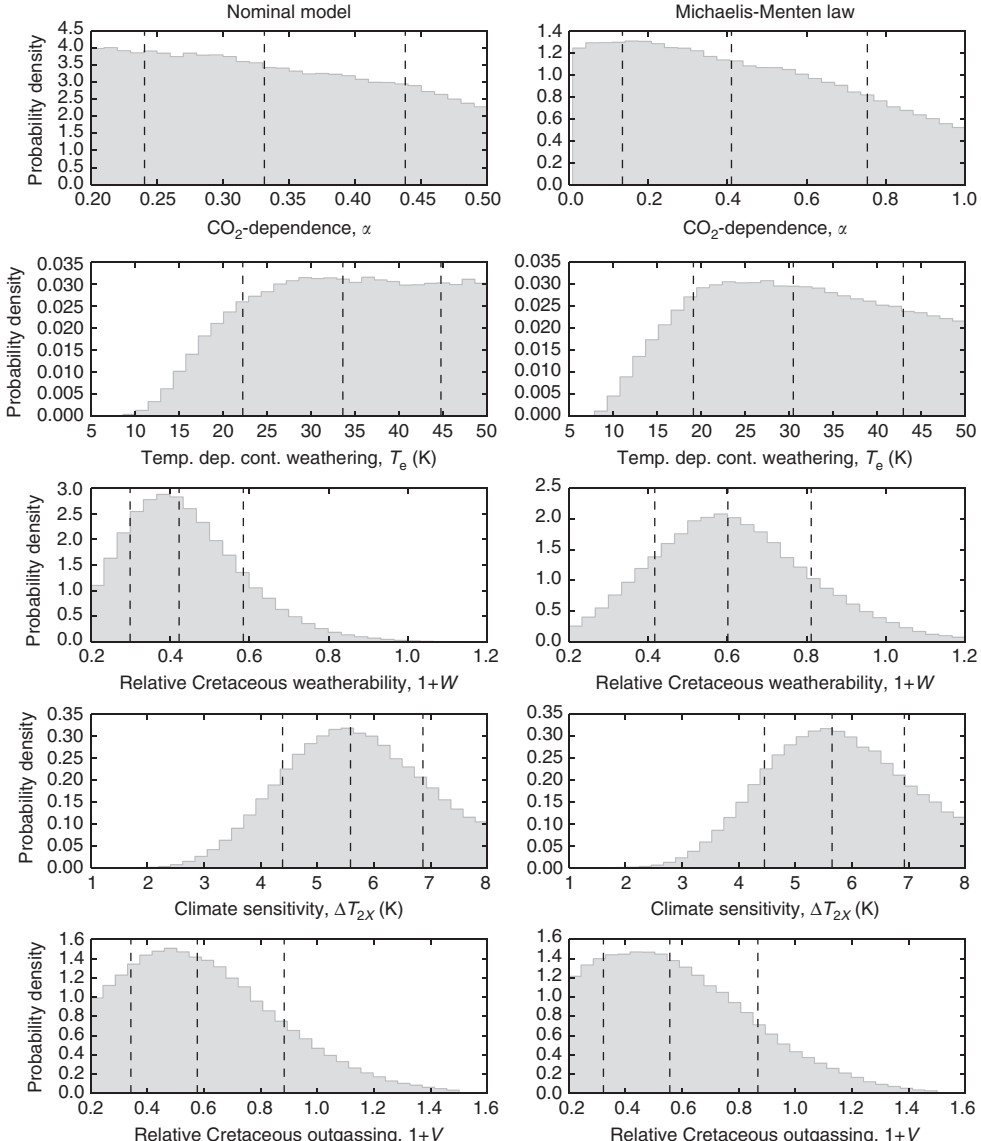

**Figure 6 | Posterior probability distributions for selected carbon cycle variables from Bayesian inverse analysis.** The first column is the posterior distributions from our nominal model, whereas the second column is distributions from a modified model, where the Michaelis–Menten law is used to describe the $pCO_2$ dependence of continental weathering. Results for the two parameterizations are similar. Dotted lines represent the median value with $1\sigma$ error bars. From the marginal distributions, we conclude that the temperature sensitivity of continental weathering is weak ($T_e > 15$ K), Cretaceous silicate weatherability ($1+W$) was ∼ half modern weatherability, and the average equilibrium climate sensitivity is ∼ 5.6 K for a $CO_2$ doubling.

ago is much larger (2.6–9.2x modern, 95% credible interval) than the relative change in continental silicate weathering (0.91–1.9x modern flux, 95% credible interval). We also observe that the continental weathering flux is probably greater than the seafloor weathering flux since 100 Myr ago (Fig. 5e,f). However, we cannot say whether the absolute change in the continental weathering sink is greater than the absolute change in the seafloor weathering sink (Fig. 5i). The carbonate weathering flux at 100 Myr ago was 18–94% the modern flux (95% credible interval, shown in Fig. 5g).

Supplementary Fig. 4 shows posterior probability distributions for variables that can only be tentatively constrained. For example, the retrieval suggests the gradient relating deep-ocean temperatures to surface temperature, $a_{grad}$ (see equation (12)), is $0.99^{+0.15}_{-0.12}$ ($1\sigma$), consistent with the linear regression in Fig. 8 (see Methods). The timescale for one ocean volume to circulate through the pore-space is ∼ 0.6 Myr, which suggests extremely short circulation times (for example, ref. 30) are unlikely, but not

excluded. The effective activation energy for seafloor basalt dissolution is probably between 53 and 97 kJ mol$^{-1}$ ($1\sigma$), in agreement with the $92 \pm 7$ kJ mol$^{-1}$ value derived by Coogan and Dosso[17], and consistent with the range reported in field studies[31]. The median value of our posterior distribution is considerably higher than experimentally derived activation energies[14], perhaps suggesting that short-term experiments do not accurately capture temperature dependence on geological timescales, although our posterior distribution is sufficiently broad that these experimental activation energies cannot be excluded. Supplementary Fig. 5 shows probability distributions for the remaining variables, which are all unconstrained (flat posterior distributions).

**Bayesian analysis sensitivity tests.** Error bars for geochemical proxies might underestimate the true uncertainties, so how robust are our results to different proxies? The large weatherability change and weak temperature dependence of continental

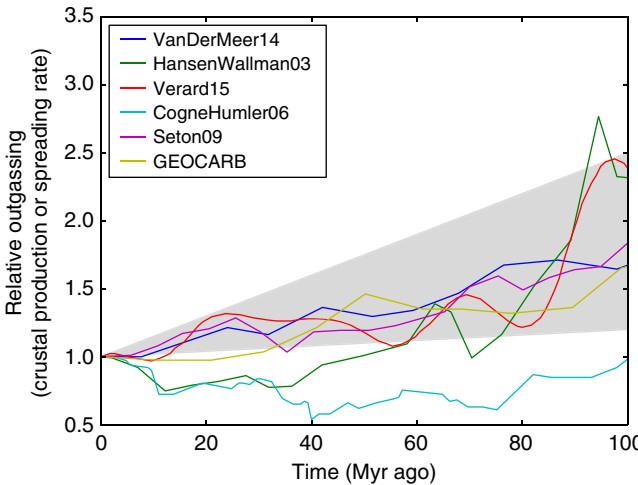

**Figure 7 | Global outgassing reconstructions.** Reconstructions of crustal production rates or spreading rates, relative to modern. These histories are assumed to reflect global outgassing histories. The studies listed in the legend are given in Supplementary Note 7. The grey region is the range of outgassing histories explored in our model.

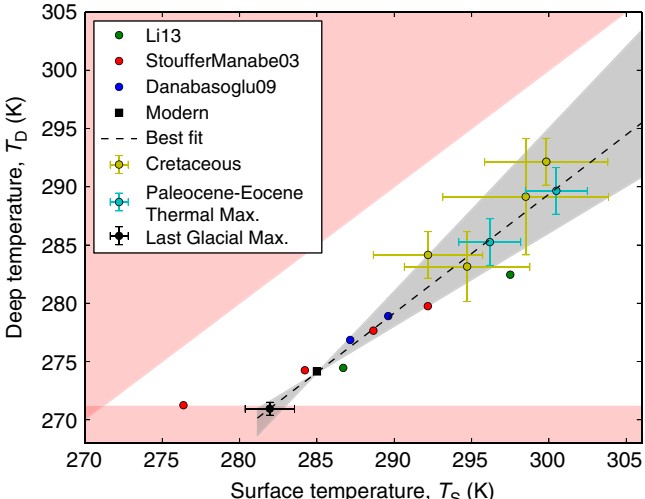

**Figure 8 | Linear relationship between global mean surface temperature and deep-ocean temperature.** Empirical relationship between deep-ocean temperatures and global mean surface temperatures, as determined by Global Circulation Model (GCM) outputs from the literature (coloured circles) and proxy data (coloured circles with error bars). The red regions are unphysical because the ocean is frozen or the deep ocean is warmer than the surface. The dotted line is the best fit to the proxies, $T_D = 1.02 T_S - 16.67$, and the grey-shaded region is the range of $T_D$-$T_S$ relationships considered in our model. See Supplementary Note 1 for full references, estimation of error bars, and further explanation.

weathering implied by our analysis do not depend on any single proxy. If the inverse analysis is repeated with the temperature, $CO_2$, pH, saturation state or seafloor weathering constraints individually omitted (equivalent to assuming we have no knowledge of these variables), then the posterior distributions for $T_e$ and $1 + W$ are largely unchanged (not shown). In fact, even if any two of those constraints are simultaneously omitted, the conclusions are unchanged. The posterior distributions for $T_e$ and $1 + W$ flatten only when three or more proxies are omitted.

In contrast, climate sensitivity results are less robust to proxies. If temperature constraints are omitted from the Bayesian analysis, then climate sensitivity is constrained to $\Delta T_{2x} = 4.5^{+2.1}_{-1.9}$ K, whereas if $CO_2$ constraints are omitted, then $\Delta T_{2x} = 3.7^{+1.5}_{-1.0}$ K. When both temperature and $CO_2$ proxy constraints are omitted, the posterior distribution for climate sensitivity becomes approximately flat. Thus, our conclusions regarding climate sensitivity are closely tied to temperature and $pCO_2$ proxies, as one might expect.

### Discussion

Four important findings emerge. First, the e-folding temperature of continental weathering lies between 17 and 48 K (90% credible) compared to the generally assumed 5–15 K. Weak temperature sensitivity of continental weathering has been suggested in previous studies of the Cenozoic and Mesozoic[32], but here we have rigorously constrained the temperature dependence.

While laboratory experiments on silicate weathering show a strong temperature dependence, it is difficult to isolate the temperature dependence in field studies because temperature covaries with other variables that modulate weathering such as precipitation, vegetation, prior soil development and cation leaching[33]. In addition, the global silicate weathering flux is mixture of transport-limited and kinetically limited regimes, and so extrapolating from field studies to a global temperature dependence is challenging. Nonetheless, a growing literature shows weak correlation between silicate weathering rates and temperature and precipitation, but strong correlation with physical erosion, which is controlled by tectonic uplift[34–37]. In addition, reactive transport modelling reveals that weathering of granitic landscapes is mostly controlled by hydrological transport, not kinetics[38]. Hydrological modelling shows that the response of global silicate weathering rates to changes in temperature depends strongly on uplift, and that the overall temperature dependence is dominated by a weak, indirect runoff dependence[8].

Riebe *et al.*[39] used cosmonucleotides and mass balance arguments to infer long-term weathering rates for 42 diverse granitic landscapes. They found silicate weathering is predominantly transport limited, with an effective activation energy of only $14–24 \, kJ \, mol^{-1}$ ($T_e = 28–48$ K), consistent with our MCMC inversion. Our results thus support the notion that weathering is predominantly transport limited.

On long timescales, runoff rates could modulate erosion, so it is challenging to isolate climatic effects[5,40]. Nonetheless, a weaker silicate weathering feedback might suggest larger swings in temperature over Earth history, though extremes will be dampened by strongly temperature-dependent seafloor weathering and the direct $CO_2$ dependence of continental weathering.

The second important finding is that silicate rock weatherability at 100 Myr ago was considerably lower than the modern. Kump and Arthur[13] inferred Cenozoic weatherability through time by forcing a carbon cycle model with crustal production, exposed land area and organic burial from proxies. They found that to fit $pCO_2$ proxies, weatherability as defined here must increase by ~0.3 over the Cenozoic, or equivalently, $1 + W = 0.53$ by linear extrapolation. In this study, the $1\sigma$ interval for $1 + W$ extends from 0.3 to 0.58. Caves *et al.*[41] also calculated the time evolution of weatherability across the Cenozoic and found a secular increase consistent with this study.

Of several possible explanations for weatherability increases, most obviously, mid-Cretaceous sea-level was 85–270 m higher than today[42]. This implies that the continental land area was 10–27% less than modern (Fig. 3.3 in ref. 43). However, given that weatherability may be weaker than linearly related to area[23] and the posterior distribution for the weatherability change (Fig. 6), it is unlikely that sea-level variation alone can account for the required change in weatherability over the last 100 Myr ago.

Weatherability can also increase with continental relief. Greater uplift may result in more physical erosion and enhanced chemical weathering[8,9]. If mid-Cretaceous relief was considerably lower than modern relief, this could explain the required change in weatherability. Berner[23] used $^{87}Sr/^{86}Sr$ ratios and terrigeneous sediment abundances to conclude that continental relief in the mid-Cretaceous was $\sim 60$–$80\%$ modern (Fig. 2.2 in ref. 23). However, the conventional interpretation of the Sr isotope record has recently been challenged[17,44]. In particular, the Cenozoic $^{87}Sr/^{86}Sr$ record can be reproduced by a simple model of temperature-dependent seafloor weathering so that no changes in continental weatherability are required[17]. Nonetheless, if these nominal changes in relief are accepted, then the combined change in weatherability from sea-level variation and reduced uplift is 0.44–0.72, which could easily explain the weatherability change inferred from our Bayesian analysis.

Changes in lithology, palaeogeography and biology also have an impact on weatherability. Basaltic weathering contributes one-third of the global silicate weathering flux despite only constituting $\sim 10\%$ of the global silicate area[45]. Reconstructions of basaltic area through time vary by methodology[46,47], but recent analyses suggest a basaltic area at 100 Myr ago approximately double the modern area[10], seemingly implying that silicate weatherability was $\sim 30\%$ greater than today. This change has opposite sign to the change implied by our retrieval, suggesting that either the increase in relief since 100 Myr ago compensated for lithology changes, or basaltic weatherability has been overestimated[48]. Palaeogeography changes are one of the primary controls on weathering for much of the Phanerozoic[49], but for the last 100 Myr ago the effect of palaeogeography on weathering fluxes has been relatively muted (Fig. 8 in ref. 49 and Fig. 4 in ref. 50). Biologically mediated changes in weatherability due to the proliferation of angiosperms[51] or the emergence of ectomycorrhizal fungi[11] could also have contributed to the weatherability increase. However, both biological innovations occurred around 100 Myr ago, suggesting that they may not be important for the relevant time span. Consequently, the most probable contributors to the required increase in weatherability over the last 100 Myr ago are lower sea levels and enhanced uplift.

Our third key finding is that average Earth system climate sensitivity since 100 Myr ago is 3.7–7.5 K for a doubling of $pCO_2$ (90% credible), which is much larger than the IPCC[28] range, 1.5–4.5 K. Our result is broadly consistent with previous estimates of Earth system climate sensitivity based on palaeoclimate data[52,53], and supports the view that the long-term climate sensitivity of the Earth system is greater than the fast-feedback Charnay sensitivity captured by Global Circulation Models (GCMs). In addition, high climate sensitivity could help explain why extremely high $pCO_2$ (for example, 4,000 p.p.m.) levels are required to reproduce observed Cretaceous equator-to-pole temperature gradients in GCMs[54].

A caveat is that we have neglected other greenhouse gases (GHGs), which could lead to overestimating climate sensitivity. Indeed, Earth system models suggest that other GHGs may have contributed significantly (2–3 K) to Cretaceous and Eocene warmth[55]. However, this increase in GHG abundances is largely a vegetation response to warmer temperatures, and GHG feedbacks such as this are already implicitly captured in our overall $pCO_2$-dependent climate sensitivity. In principle, the omission of methane forcings could affect our results, but the changes in methane flux would have to be substantial to reduce our inferred climate sensitivity. The warming from other GHGs is only $\sim 0.5$ K from changes in boundary conditions between the Early Eocene and Late Cretaceous[55].

Fourth, although the continental weathering flux was probably larger than the seafloor weathering flux since 100 Myr ago

(Fig. 5e,f), it is difficult to directly compare the importance of the two feedbacks. The temperature dependence of continental weathering is likely weak, but both the temperature dependence of seafloor weathering and the direct $pCO_2$ dependence of continental weathering are poorly constrained (Fig. 6; Supplementary Fig. 4). Consequently, we cannot say whether the absolute change in continental weathering since 100 Myr ago is greater or less than the change in seafloor weathering (Fig. 5i). In addition, conclusions regarding low-temperature sensitivity of continental weathering, weatherability changes and climate sensitivity hold irrespective of assumptions about seafloor weathering; these conclusions arise primarily from the fit to temperature, $pCO_2$, saturation state and pH proxies.

With that said, the temperature dependence of continental weathering ($E_a = 20^{+10}_{-5}$ kJ mol$^{-1}$) is weak compared to seafloor weathering ($E_{bas} = 75^{+22}_{-21}$ kJ mol$^{-1}$), and a secular decline in spreading rates over Earth history would imply more dominant seafloor weathering fluxes at earlier times since continental weathering has no direct spreading rate dependence. Taken together, these observations suggest that seafloor weathering feedback may have been an important feedback at earlier times in Earth's history.

All of our conclusions—to varying degrees—depend on the fidelity of proxies. We have attempted to minimize this source of error by adopting the broadest possible range of proxy estimates for $pCO_2$, temperature, saturation state, pH and the seafloor weathering sink (see Methods). The sensitivity analysis in the results section shows that the low-temperature sensitivity of continental weathering and the large weatherability increase since 100 Myr ago are robust. Even if current estimates of two proxies are highly uncertain or flawed, the remaining proxies tell a mutually consistent story on continental weathering. However, the same is not true for climate sensitivity. If proxies overestimate global mean temperatures in the Cretaceous, then Earth system climate sensitivity may be lower than our inverse analysis suggests. The same is true if real Cretaceous $pCO_2$ was much higher than proxy estimates.

To some extent, conclusions are sensitive to the mechanistic assumptions in our forward model. For example, the simple functional forms adopted for many biogeochemical fluxes and the reduction of spatially heterogeneous processes to zeroth order, globally averaged equations could influence our results. However, wherever possible, we used parameterizations that have a fundamental physical basis, such as the logarithmic dependence of climate on $pCO_2$ and the Arrhenius-style temperature dependence of weathering. In instances where the physical basis is uncertain, we adopted generalized power laws with widely varying exponents to describe relationships between variables, and introduced free parameters to account for unknown processes. Column 2 of Fig. 6, and Supplementary Note 6 explore the sensitivity of our results to different continental weathering parameterizations. We find that none of the key conclusions are changed by using different functional forms. Nonetheless, our quantitative estimates of key variables could be refined by better mechanistic understanding carbon cycle relationships. Specifically, the magnitude of the weatherability increase since 100 Myr ago is moderately sensitive to the choice of continental weathering function (Fig. 6; Supplementary Note 6).

Another potential limitation is that we are imposing linear trends in some model parameters, thereby potentially underfitting the data and underestimating the uncertainties in retrieved parameters. In Supplementary Note 5, we repeat our analysis with a simplified data set to show that our conclusions are robust to these linearity assumptions. Supplementary Note 4 also shows that possible changes in K-feldspar uptake[16] in the seafloor do not change our qualitative conclusions.

In summary, we presented a new geological carbon cycle model, which includes ocean chemistry and the kinetics of seafloor weathering, and applied it to the last 100 Myr ago. Model outputs were compared to proxies for temperature, atmospheric $CO_2$, seafloor carbonate content, ocean pH and ocean saturation state. A MCMC inversion rigorously constrained carbon cycle parameters given these data. Assuming that proxies are accurate and that our forward model accurately parameterizes the carbon cycle, we report five key conclusions. First, the temperature dependence of continental silicate weathering is considerably weaker than commonly assumed. Most carbon cycle models use an effective activation energy to $\sim$50–100 kJ mol$^{-1}$, whereas our results imply $E_a = 20^{+10}_{-5}$ kJ mol$^{-1}$, which suggests that continental silicate weathering is less effective at buffering climate against changes in outgassing or insolation. Second, mid-Cretaceous continental weatherability was 30–58% of modern weatherability (1$\sigma$), although the precise magnitude of the change is sensitive to the functional form adopted for continental weathering. This increase in weatherability since 100 Myr ago is best explained by continental uplift and sea-level decline. Third, the average Earth system climate sensitivity is $\Delta T_{2x} = 5.6^{+1.3}_{-1.2}$ K for a $CO_2$ doubling (1$\sigma$). This is considerably higher than fast-feedback estimates for the modern climate, and could explain why extremely high p$CO_2$ levels are required to reproduce greenhouse climates in GCMs. This result is derived assuming methane variations are fully captured by temperature-dependent feedbacks, and it is sensitive to uncertain temperature proxies. Fourth, Cretaceous outgassing was unlikely ($\sim$9% probability) to be greater than double modern outgassing. Finally, continental weathering is probably the dominant carbon sink throughout the last 100 Myr ago, and introducing seafloor weathering into our model has a relatively small effect on the inverse modelling results. However, the strong temperature sensitivity and spreading rate dependence of seafloor weathering implies that it could have been a dominant carbon sink earlier in Earth history.

## Methods

**Model description.** Python code for the model is available open source from the first author's website. The time evolution of the carbon cycle is described by the following set of equations:

$$\frac{dC_O}{dt} = \frac{-J(C_O - C_P)}{M_O} + \frac{F_{out}}{M_O} + \frac{F_{carb}}{M_O} - \frac{P_{ocean}}{M_O}$$

$$\frac{dA_O}{dt} = \frac{-J(A_O - A_P)}{M_O} + 2\frac{F_{sil}}{M_O} + 2\frac{F_{carb}}{M_O} - 2\frac{P_{ocean}}{M_O}$$

$$\frac{dC_P}{dt} = \frac{J(C_O - C_P)}{M_P} - \frac{P_{pore}}{M_P} \quad (6)$$

$$\frac{dA_P}{dt} = \frac{J(A_O - A_P)}{M_P} + 2\frac{F_{diss}}{M_P} - 2\frac{P_{pore}}{M_P}$$

Here $C_O$ and $C_P$ are the concentrations of carbon (Tmol C kg$^{-1}$) in the atmosphere–ocean and pore-space, respectively. The carbon concentration in the pore-space is equivalent to the dissolved inorganic carbon (DIC) abundance, $C_P = DIC_P$, whereas carbon in the atmosphere–ocean reservoir is equal to marine DIC plus atmospheric carbon, $C_O = DIC_O + pCO_2 \times s$, where $s$ is a scaling factor equal to the ratio of total number of moles in the atmosphere divided by the mass of the ocean, $s = (1.8 \times 10^{20})/M_O$. Similarly, $A_O = ALK_O$ and $A_P = ALK_P$ are the carbonate alkalinities in the atmosphere–ocean and pore-space, respectively (Tmol eq kg$^{-1}$). The global outgassing flux (Tmol C per year) is specified by $F_{out}$, whereas the rates of continental silicate weathering and carbonate weathering are $F_{sil}$ and $F_{carb}$, respectively (Tmol C per year). Seafloor weathering from basalt dissolution (Tmol eq per year) is $F_{diss}$, and the precipitation flux of carbonates (Tmol C per year) in the ocean and pore-space are given by $P_{ocean}$ and $P_{pore}$, respectively. The mass of the ocean and the pore space is given by $M_O = 1.35 \times 10^{21}$ kg and $M_P = 1.35 \times 10^{19}$ kg, respectively[20]. We assume a range for $J$ of $1.4 \times 10^{15} - 6.8 \times 10^{16}$ kg per year, which implies the time to circulate one ocean volume through the pore-space is between $\tau = 20$ and $\tau = 1,000$ kyr, consistent with estimates from Johnson and Pruis[56] and Caldeira[20].

A common simplification in carbon cycle modelling is to neglect carbonate weathering[15,22,24]. This is justified on the grounds that carbonate weathering does not constitute a net carbon source. On long timescales, the carbon consumption

from silicate and seafloor weathering must balance carbon outgassing (plus any imbalance in organic weathering and burial), and this balance determines atmospheric $CO_2$ and climate. However, the saturation state of the ocean is affected by carbonate weathering, and so we include carbonate weathering to model ocean chemistry. We do not track crustal or mantle reservoirs, and the atmospheric reservoir of $CO_2$ is set by equilibrium partitioning with the ocean (see below). Because we are not tracking fluxes between the atmosphere and ocean, continental silicate weathering is not a carbon source or sink; the release of cations (alkalinity) from silicate dissolution does not directly add carbon to the combined atmosphere–ocean system. Instead, the cations consume carbon indirectly when they later precipitate as marine carbonates.

There is no surface ocean box in our model. This is because we are only interested in the changes in properties of the bulk ocean, which are determined by varying boundary conditions not by the deep-surface partitioning. In addition, we are focused on timescales $>10^6$ years, and so the dynamics of deep-surface ocean mixing are unimportant. If we were to partition the ocean into surface and deep, the main difference would be differential temperature dependence of carbon speciation. We also do not include organic carbon weathering and burial. This is justifiable because the negative feedback from oxidative weathering ensures they are approximately balanced[57]. Empirically, the carbon isotope record reveals that the organic burial fraction has changed by only $\sim$10% over the last 100 Myr ago[58]. Consequently, the change in the organic burial flux is likely modest and so the omission of organic carbon will not affect our conclusions.

The functional forms for the flux terms not already described in the main text are presented below.

**Continental silicate weathering.** The continental weathering parameterization was described in the main text except for the coefficient $\alpha$ in equation (2). This coefficient is assumed to be 0.2–0.5 (ref. 59). Strictly speaking, soil p$CO_2$ should replace atmospheric p$CO_2$ in equation (2) because soil p$CO_2$ determines soil pH, and therefore silicate dissolution rates. However, if modest changes in maximum biological productivity are allowed, then the range of $CO_2$ dependencies from the term $\left(pCO_2/pCO_2^{mod}\right)^\alpha$, with $\alpha$ varying from 0.2 to 0.5, is broadly equivalent to replacing atmospheric p$CO_2$ with soil p$CO_2$ (ref. 60; see Supplementary Fig. 6). Consequently, we retain atmospheric p$CO_2$ and $pCO_2^{mod}$ in equation (2). To test the sensitivity of our results to weathering parameterizations, in addition to the p$CO_2$ power-law dependence in equation (2), we also consider a Michaelis–Menten law in the main text (see results).

The range adopted for relative Cretaceous weatherability, $1 + W$, is based on literature estimates[10,23,43,47,49] of how external factors may have affected weatherability, and extending the range in either direction does not markedly change our results. The weatherability factor, $\omega$, can also be interpreted as the sensitivity of the weathering response to changes in p$CO_2$ and temperature. For example, an increase in $\omega$ implies that an increase in surface temperature will result in a larger change in the continental weathering flux, $F_{sil}$.

**Continental carbonate weathering.** We assume that carbonate weathering has the same functional form as silicate weathering, except for an additional dimensionless multiplicative factor, $\omega_{carb}$, to allow for the possibility that carbonate weathering is subject to different temperature dependence, $CO_2$ dependence and weatherability factors:

$$F_{carb} = \omega_{carb} \, \omega F_{carb}^{mod} \left(\frac{pCO_2}{pCO_2^{mod}}\right)^\alpha \exp(\Delta T_S/T_e) \quad (7)$$

The carbonate weatherability factor is defined as follows:

$$\omega_{carb} = (1 + C_{WF}t/100 \, \text{Myr}) \quad (8)$$

We assume a range of values for $C_{WF}$ from $-0.9$ to 1.5 to allow for large differences between carbonate weathering and silicate weathering. For example, carbonates may have a lower effective activation energy compared to silicates because carbonate weathering is sensitive to runoff, whereas silicate weathering is sensitive to both runoff and a kinetic temperature effect[7,25]. Carbonate weathering may also have a different response to changes in uplift[23], or varying fluxes due to changes in the crustal reservoir of carbonates. It should be noted that changes in carbonate weathering only affect saturation state; the changes in $CO_2$, temperature and ocean pH due to carbonate weathering changes are negligible. This is because—as explained above—temperature and $CO_2$ are set by the balance between outgassing and silicate weathering plus seafloor weathering. Consequently, any conclusions drawn about those variables are unaffected by our formulation for carbonate weathering.

**Climate model.** To relate $\Delta T_S$ to p$CO_2$, we adopt the following climate model:

$$\Delta T_S = \Delta T_{2x}\left(\frac{ln(pCO_2/pCO_2^{mod})}{ln(2)} - \frac{t}{228 \, \text{Myr}}\right) + \Delta P\left(\frac{t}{100 \, \text{Myr}}\right) \quad (9)$$

Here $\Delta T_{2x}$ is the climate sensitivity parameter, $\Delta P$ is a palaeogeography parameter and the second term accounts for solar luminosity changes (Supplementary Note 2). We divide by ln(2) so that $\Delta T_{2x}$ has conventional units of Kelvin warming per $CO_2$

doubling. Supplementary Fig. 9 compares different climate parameterizations and GCM results from the literature and illustrates why equation (9) is suitable. The parameter $\Delta P$ is the secular cooling (in K) since the mid-Cretaceous due to palaeogeography changes. A review of GCM studies concluded that $\Delta P = 0–3.0$ K (ref. 52). We assume $\Delta P = 0–5.0$ K to be conservative, noting that some models suggest 5 K of warming from an Eocene continental configuration[61].

**Outgassing.** Estimates of Cenozoic and Mesozoic outgassing histories vary substantially. Figure 7 shows a variety of outgassing reconstructions from the literature expressed as crustal production or spreading rates, which are assumed to covary with global outgassing. Generally, these reconstructions suggest that global outgassing at 100 Ma was between $1.5x$ and $2.5x$ modern outgassing. This conclusion is based on several independent lines of evidence including reconstructions of plate extent and plate motion, seismic imaging of subducted plates, and reconstructions of seafloor age and depth (see Supplementary Note 7 for a summary of outgassing estimates with references). The outlying reconstruction in Figure 7 (aqua curve) is disputed because it uses a contentious crustal age distribution[42].

For simplicity, we assume a linear global outgassing history:

$$F_{out} = F_{out}^{mod}(1 + Vt/100 \text{ Myr}) \tag{10}$$

Here $F_{out}^{mod}$ is modern outgassing (Tmol C per year), $t$ is time (in Myr ago) and $V$ is a dimensionless scaling factor. Using crustal production or spreading rate as a proxy for global outgassing is a simplification because it ignores subaerial metamorphism and hot spot volcanism (for example, ref. 62). Given the uncertainty in these other contributions, we adopt a very broad range of outgassing histories since 100 Myr ago by assuming $V = 0.2–1.5$. Thus, we allow mid-Cretaceous outgassing to range from 20 to 150% greater than modern.

**Basalt dissolution and seafloor weathering.** The temperature dependence of seafloor weathering uses the following Arrhenius-style expression[17]:

$$F_{diss} \propto \exp(-E_{bas}/RT_{pore}) \tag{11}$$

Here $E_{bas}$ (kJ mol$^{-1}$) is the effective activation energy of basalt dissolution, $R$ is the universal gas constant and $T_{pore}$ is the pore-space temperature. Coogan and Dosso[17] reported an empirically derived activation energy of $E_{bas} = 92 \pm 7$ kJ mol$^{-1}$, whereas experimental studies[14,31] of basalt dissolution suggest activation energies between 42 and 109 kJ mol$^{-1}$. We adopt a range of activation energies from $E_{bas} = 40$ to 110 kJ mol$^{-1}$.

Because Cenozoic and Mesozoic pore-space temperatures are controlled by deep-ocean temperature[17], we must determine the link between global mean surface temperatures and deep-ocean temperatures. Figure 8 shows mean global surface temperatures plotted against deep-ocean temperatures using output from fully coupled atmosphere–ocean GCMs and palaeoclimate proxy data (see Supplementary Note 1 for details). The relationship is described by an empirical linear fit:

$$T_D = a_{grad}T_S + b_{int} \tag{12}$$

Here $T_D$ (K) is the mean deep-ocean temperature and $T_S$ (K) is the mean surface temperature. The best-fit gradient and intercept are $a_{grad} = 1.02$ and $b_{int} = -16.7$, respectively. However, we assume a broad gradient range $a_{grad} = 0.8–1.4$, whereas the intercept, $b_{int} = 274.037 - a_{grad} \times 285$ is chosen to ensure consistency with modern conditions. Figure 8 shows the range of possible $T_D \propto T_S$ relationships used in this study.

Our parameterization of deep-ocean temperature improves upon Brady and Gíslason[14] because ours is based on an ensemble of GCM results and globally averaged palaeoclimate proxies rather than a single climate model. In addition, Brady and Gíslason[14] overestimated the dependence of deep-ocean temperature on surface climate because their parameterization is based on a single near-equatorial latitude of 6.7°. The relationship between globally averaged abyssal temperatures and surface climate is more muted than the relationship with equatorial abyssal temperatures. This can be seen in our Fig. 8 and in Fig. 12 of Manabe and Bryan[63], despite the model of the latter being the basis for the Brady and Gíslason[14] parameterization.

To relate the pore-space temperature to the deep-ocean temperature, we adopt empirical results[17]. Oxygen isotopes indicate that for both the Cenozoic and Mesozoic, the mean pore-space temperature of seafloor carbonate precipitation is consistently ~9 K warmer than the minimum temperature of seafloor carbonate precipitation (deep-ocean temperatures). Consequently, we assume $T_{pore} = T_D + 9$. This modification has a very minor effect on the model output because it is largely the change in temperature, and not its absolute value, that controls variations in the seafloor weathering flux.

Quantifying the pH dependence of seafloor weathering is more challenging because most experiments either fail to separate the pH and direct $CO_2$ effect[14], focus on individual minerals rather than whole-rock dissolution rates[64,65], or do not explore the full pH range relevant to seafloor weathering[66]. Gudbrandsson et al.[67] measure whole-rock crystalline basalt dissolution rates for $2 < pH < 11$. They find that Ca release is a U-shape function of pH (their Fig. 8), where the minimum of the 'U' at 25 °C is somewhere between pH = 7 and pH = 9, depending on the assumptions made about the reactive surface area (experimental

results are sparse and so are fitted with an analytic model of dissolution). Given the uncertainty in this dissolution curve, it is difficult to predict the sign of the dissolution change for a modest change in ocean pH. For example, a decline in pH from 8.2 to 7.4—which is approximately the change from the modern ocean to mid-Cretaceous—predicts a 20% decrease in Ca release according to one Gudbrandsson et al.[67], their Fig. 8f fit, and a 7% increase in Ca release according to the alternative Gudbrandsson et al.[67], their Fig. 8e fit. Either way, the change in dissolution is minor, and so a possible first order approximation is to assume basalt dissolution on the seafloor is independent of pH for the Mesozoic and Cenozoic.

However, the Gudbrandsson et al.[67] experiments may not accurately capture the pH dependence of seafloor weathering because they were not done in seawater and did not include carbon chemistry. Some experimental studies[65,68] show that olivine dissolution is $CO_2$-dependent at high pH values, ostensibly because abundant carbonate ions protect Si–O bonds, thereby decreasing dissolution with increasing DIC. In contrast, Golubev et al.[64] studied the effect of $CO_2$ on dissolution rates for a range of pH values and found that forsterite, diopside and hornblende do not have $CO_2$-dependent dissolution rates at any pH. Unfortunately, however, Golubev et al.[64] did not consider plagioclase, so their results cannot easily be extrapolated to the basaltic seafloor. Wolff-Boenisch et al.[66] showed that the $CO_2$ dependence of crystalline basalt is independent of $CO_2$ at low pH levels, but did not repeat the experiment at high pH values.

We allow for the possibility of pH dependence by setting the rate of dissolution proportional to $[H^+]_P^\gamma$, where $\gamma$ varies from 0 (no pH dependence) to 0.5 (strong pH dependence dominated by basaltic glass dissolution):

$$F_{diss} = k_{diss}\left(\frac{F_{out}}{F_{out}^{mod}}\right)^\beta \exp(-E_{bas}/RT_{pore})\left(\frac{[H^+]_P}{[H^+]_P^{mod}}\right)^\gamma \tag{13}$$

Here $k_{diss}$ is a proportionality constant chosen to match the modern flux, $[H^+]_P$ is the hydrogen ion molality in the pore-space and $[H^+]_P^{mod}$ is the modern molality. Because dissolution is dependent on crustal production and crustal production is proportional to global outgassing, we assume dissolution is dependent on outgassing with some unknown power–law relationship, defined by $\beta = 0–1$.

Better knowledge of the pH dependence, temperature dependence and $CO_2$ dependence of basalt dissolution would improve our constraints on the carbon cycle. Specifically, whole-rock dissolution experiments performed at high pH that separate the effects of pH and DIC would allow for more precise parameterizations of seafloor weathering.

**Precipitation fluxes.** The precipitation flux of marine carbonates, $P_{ocean}$, is the sum of the fluxes of shelf carbonates, $P_{shelf}$, and pelagic carbonates, $P_{pelagic}$:

$$P_{ocean} = P_{shelf} + P_{pelagic} \tag{14}$$

Following Ridgwell[69], the shelf precipitation flux is given by:

$$P_{shelf} = k_{shelf}\frac{A_{shelf}}{A_{shelf}^{mod}}(\Omega_O - 1)^n \tag{15}$$

Here $\Omega_O$ is the saturation state of the ocean (defined below by equation (20)), $A_{shelf}$ is the area of continental shelf available for carbonate precipitation, with $A_{shelf}^{mod}$ denoting the modern shelf area, and $k_{shelf}$ is a proportionality constant. Shelf area is approximated by a polynomial fit (Supplementary Fig. 7) to reconstructed tropical shelf area from Walker et al.[70]. The exponent $n$ defines the proportionality between the saturation state of the ocean and the precipitation flux. This is typically taken to be 1.7 based on the latitudinal dependence of carbonate accumulation and saturation state[71]. Rather than consider calcite and aragonite precipitation separately, we instead allow $n$ to vary widely from 1.0 to 2.5.

The pelagic carbonate flux depends on the calcite compensation depth (CCD), $Z_{CCD}$ (km), which can be calculated using the following equation[72]:

$$Z_{CCD} = 4 + 6.25\ln(\Omega_O) \tag{16}$$

Pelagic carbonate deposition is proportional to the fractional area above the CCD, $f(Z_{CCD})$, which can be approximated by an exponential fit to hypsometric data[73]:

$$P_{pelagic} \propto f(Z_{CCD}) = 0.07\exp(Z_{CCD}/2.2) \tag{17}$$

Equations (14–17) can then be combined to give the total ocean precipitation flux:

$$P_{ocean} = k_{shelf}\frac{A_{shelf}}{A_{shelf}^{mod}}(\Omega_O - 1)^n + k_{pelagic}\Omega_O^{2.84} \tag{18}$$

The proportionality constants $k_{shelf}$ and $k_{pelagic}$ are chosen to reproduce the modern partitioning between shelf and pelagic carbonates (see below). Here the precise functional form of equation (18) only matters for determining ocean saturation state; carbon fluxes, pCO$_2$ and temperatures are unaffected.

The pore-space carbonate precipitation flux is analogous to shelf precipitation except that there is no area dependence:

$$P_{pore} = k_{pore}(\Omega_P - 1)^n \tag{19}$$

Here $\Omega_P$ is the saturation state of the pore-space and $k_{pore}$ is a proportionality constant chosen to reproduce the modern flux. The exponent $n$ is the same as for shelf precipitation. Repeating the inverse analysis allowing different exponents for

shelf and seafloor carbonate precipitation does not change results substantially (not shown).

Finally, the saturation state of the ocean and the pore-space are defined as follows:

$$\Omega_O = \frac{[Ca^{2+}][CO_3^{2-}]_O}{K_{sp}} \text{ and } \Omega_P = \frac{[Ca^{2+}][CO_3^{2-}]_P}{K_{sp}} \quad (20)$$

Here $K_{sp} = K_{sp}(T)$ is the temperature-dependent solubility product from Pilson[74]. Supplementary Methods explain how the solubility product is calculated.

**Ocean chemistry.** Alkalinity and DIC have the following standard definitions in our model, where ALK is often referred to as 'carbonate alkalinity' in the literature:

$$DIC = [CO_3^{2-}] + [HCO_3^-] + [CO_2aq]$$
$$ALK = 2[CO_3^{2-}] + [HCO_3^-] \quad (21)$$

Given carbon and alkalinity in the atmosphere–ocean ($C_O$, $ALK_O$) or the pore-space ($C_P$, $ALK_P$), we can calculate ocean chemistry using the following set of equations[74]:

$$[CO_2aq] = pCO_2 \times H_{CO_2} \quad (22)$$

$$[HCO_3^-] = \frac{[CO_2aq] \times K_1^*}{[H^+]} \quad (23)$$

$$[CO_3^{2-}] = \frac{[HCO_3^-] \times K_2^*}{[H^+]} \quad (24)$$

$$\frac{ALK}{K_1^* K_2^*}\left(1 + \frac{s}{H_{CO_2}}\right)[H^+]^2 + \frac{(ALK-C)}{K_2^*}[H^+] + (ALK - 2C) = 0 \quad (25)$$

$$pH = -\log_{10}([H^+]) \quad (26)$$

Here $H_{CO_2}$ is the Henry's law constant for $CO_2$, $[CO_2aq]$ is the sum of the concentrations of free $CO_2$ and $H_2CO_3$, and $K_1^*$ and $K_2^*$ are the first and second apparent dissociation constants of carbonic acid, respectively. Temperature-dependent expressions for these constants can be found in Supplementary Methods. The set of equations described above must be solved separately for the ocean and the pore-space by substituting the generic carbon concentration and alkalinity ($C$, ALK) for ($C_O$, $ALK_O$) and ($C_P$, $ALK_P$), respectively. The scaling factor, $s$, is defined above with respect to equation (6). Equation (25) is derived by combining equations (21), (23) and (24) (Supplementary Methods). This quadratic can be solved to find $[H^+]$. Once this is known, then equations (22), (23) and (24) define the remaining carbon chemistry variables.

Rather than attempt to model the complexities of calcium and magnesium cycling in our model, we impose observed changes in $[Ca^{2+}]$ abundances from seawater inclusions[75]. In our model, changes in alkalinity are driven purely by weathering and carbonate precipitation. Thus, by imposing $[Ca^{2+}]$ variations, we are effectively assuming that observed $[Ca^{2+}]$ changes are offset by changes in other cations and anions, such that they have no direct effect on alkalinity, for example, magnesium exchange with the seafloor. We fit a third-order polynomial to the $[Ca^{2+}]$ reconstruction in Tyrrell and Zeebe[75] to achieve the fast computation times necessary for Bayesian inversion (Supplementary Fig. 8).

**Initial conditions and numerical solution.** Table 2 shows all the initial values assumed in our model or ranges for variables that are uncertain. All other initial values are fully determined by the variables in this table.

The system of differential equations describing the carbon cycle (equation (6)) was solved in Python using the ordinary differential equation integrator in the *SciPy* module. Model outputs were compared with equivalent steady-state calculations and were always in agreement to within a few per cent or better (Supplementary Note 3). This validates the numerical integration and implies that the time-dependent model is always in quasi steady state. In addition, the integrated flux imbalance over 100 Myr ago equals the change in the carbon reservoirs to within ∼2% or better in every case, confirming that mass is being conserved in our model.

**Proxies.** The geochemical proxies plotted in Figs 2–5, and Supplementary Figs 2, 15 and 16 are described in Supplementary Methods. For each variable, we searched the literature for the broadest possible range of proxy estimates. Proxies were typically binned into 10 Myr ago intervals, and for each interval the best estimate was taken to be the midpoint of the full range of proxy estimates, while the $1\sigma$ uncertainty in the best estimate spanned the full range (Supplementary Figs 10–13). This conservative approach helps ensure that our conclusions are robust to uncertainties in different proxy methods.

**Data availability.** The binned proxy data used as inputs for this analysis along with the carbon cycle model code are available on the website of the first author. www.krisstott.com

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

## Acknowledgements

We thank Dorian Abbott, Lee Kump and the two anonymous reviewers whose comments greatly improved the manuscript. We also thank Giada Arney, Laurence Coogan, James Kasting, Rodrigo Luger and Chris Reinhard for helpful discussions. This work was supported by NASA Exobiology Program grant NNX15AL23G awarded to D.C.C. and by the NASA Astrobiology Institute's Virtual Planetary Laboratory, grant NNA13AA93A. J.K.-T. is supported by NASA Headquarters under the NASA Earth and Space Science Fellowship program, grant NNX15AR63H.

## Author contributions

Both authors contributed to the conception of this project and the drafting of the manuscript. J.K.-T. created the model and performed the analysis.

## Additional information

**Competing interests:** The authors declare no competing financial interests.

