## [Peer Review File · Nature Communications]

Reviewers' comments:

Reviewer #1 (Remarks to the Authors)

Paper: The global carbon cycle since the mid Cretaceous: Implications for climate sensitivity and continental versus seafloor weathering

Authors: Krissansen-Totton and Catling

Reviewer: Dorian S. Abbot

Date: July 1, 2016

Overview: The authors build a new carbon cycle model and perform a Bayesian statistical analysis using it that allows them to infer parameter values of their model based on other people's observations. Their most interesting conclusions are that continental weathering responds much more weakly to temperature than we had previously realized, and that seafloor weathering does seem to respond to the surface climate, allowing a negative climate-seafloor weathering feedback. The paper is clear and the arguments are well-presented. The work is impressive and brings together a huge amount of data to make relevant and exciting inferences. I consider this work significantly more sophisticated than what is typically seen in paleoclimate/paleoceanography papers. I think this paper deserves quick publication and will have a large impact on the field. I give a few comments for improvement below.

Comments:

1. Climate Sensitivity: I am not convinced by the climate sensitivity inference for a number of reasons. The first is just that we do not have any information about other greenhouse gasses such as methane (and we have reason to suspect that they might be higher in a hot-house climate). Neglecting these leads to a big upward bias in the inferred climate sensitivity.

Also, I was not able to find a description of the CO₂ proxy data. Maybe I just missed it, or maybe it could be in a more prominent place. Anyway, I do not see error bars on the data points from the Cenozoic when these are plotted in the B panels of the relevant figures. What are you assuming for the errors when you use these in the Bayesian analysis? Also, are these from leaf stomata? That method is notoriously unreliable, especially when trying to infer high CO₂ values because the method saturates. If the CO₂ really were higher during the Eocene than you think, the climate sensitivity you infer would be lower than it should be.

Finally, I am not satisfied with the claim that changes in insolation roughly balance changes in radiative forcing due to continental configuration. There are two major issues with this: (1) the two effects have different patterns in time and there's way they perfectly canceled over the entire 100 Myr period considered and (2) the radiative forcing due to changing the continental configuration is going to be model dependent, mainly due to cloud effects (just because one model says it cancels with insolation does not mean this is true when you "marginalize" over reasonable models). For example, consider the Eocene. If we assume a planetary albedo of 0.3, then the solar forcing was something about 1-1.5 W m⁻² lower during the Eocene. If we assume a climate sensitivity of 0.8 K W⁻¹ m², we get about 1 K of cooling due to the lower solar forcing. In CCSM3, going to Eocene configuration causes 5 K of warming (*Caballero and Huber, 2013*), although it's a little unclear how much of that to count as forcing and how much as Earth system sensitivity. Anyway, if you think the Eocene CO₂ is low because of (questionable) leaf stomata proxy data, you leave out 4 K of net warming due to changes in forcing, and you do not consider forcing by other greenhouse gasses, you are going to infer a climate sensitivity that is way too high!

The bottom line is that I would try a little harder on the CO₂ proxy error bars, I would try to include variations in solar forcing and forcing due to changes in continental configuration, and I would play down the claim to have inferred climate sensitivity since we have no information on other greenhouse gasses (or maybe call the inference you make an upper limit).

2. Importance of weathering: The work you have done gives you the chance to do some cool calculations of the importance of weathering feedbacks. For example, you could run the forward model with all weathering feedbacks turned off and see how the climate evolves. You could also switch off either the continental or seafloor weathering. This would be a nice way to show which one is more important. I think this would help make the paper accessible to the paleoclimate crowd, who may get lost in the Bayesian analysis.

3. Details: Figure 6 is supposed to contain posterior distributions of well-constrained variables. α , T_e , and $1 + V$ do not seem very well-constrained, in that the posterior does not seem to taper to low values at their extremes. Would it make sense to try larger ranges in these variables so that you sample values that the data can exclude?

Why not present the temperature dependence of seafloor and continental weathering using the same units so they can be compared more easily?

You note that continental weathering dominates over seafloor weathering, but it seems that the changes in these variables are more important than their total magnitudes if we care about a negative weathering feedback. It's a little hard to tell from Figure 5E,F which is changing more because of the difference in the scales, but this would be worth discussing.

References

Caballero, R., and M. Huber (2013), State-dependent climate sensitivity in past warm climates and its implications for future climate projections, *Proceedings of the National Academy of Sciences*, *110*(35), 14,162–14,167.

Reviewer #2 (Remarks to the Author):

In this paper, the authors apply a new model to evaluate the mechanisms operating to steer changes in the global carbon cycle over the past 100 million years. One significant addition relative to prior studies on this topic comes in the way that the authors treat seafloor weathering. Though not totally ignored in previous work, this component of the carbon cycle has not had the treatment that it deserves (or needs), so this aspect of the authors' contribution is a particularly welcome addition. The authors conclude that seafloor weathering has played a seminal role in charting the course of C cycle evolution, more than previously thought, and that the responsiveness of terrestrial weathering to changes in climate (the "negative weathering feedback" that stabilizes long-term climate) has been over-estimated in past work. The authors' results also suggest that global "weatherability" has changed over the past 100 Ma, confirming inferences made in a number of prior studies. Nonetheless, as far as I can tell this study can only weakly constrain the magnitude of change in weatherability and cannot rule out equally or more important changes in outgassing, in terms of contributing to the overall cooling trend since the Cretaceous. In other words, the authors are not able in this study alone to answer the "uplift vs. degassing" debate, for what that duality is worth. In any case, the authors here are more concerned about understanding how feedbacks have operated to maintain a stable climate system. Their conclusions in these regards are without a doubt interesting and novel, i.e., about relative roles of seafloor vs. terrestrial weathering, and about the strength of climate sensitivity and weathering feedbacks. Overall, at least from my perspective, these are important questions, so this promises to be a widely read study on a topic of significant interest in the geosciences, in an area where a number of questions remain to be fully resolved. And in addition to the provocative interpretations, I am confident that the paper provides some new modeling directions that may help take the field forward.

On the other hand, in my view a substantial weakness of this paper is that the authors arrive at a number of conclusive statements that I am not sure are fully supported by their analysis. Or at least I think the authors are more definitive in the way that they present their conclusions than seems really warranted to me. The authors use a Bayesian inversion to arrive at a set of parameters that they argue best describe the evolution of the C cycle. Because their model arrives at a best-fit solution, they conclude that the best-fit parameters must describe the actual system behavior. While I agree that this kind of inverse approach has a lot to offer in addressing the problem at hand, I think the paper in its current form is at risk of being overly conclusive, principally because the model itself depends on poorly known inputs.

A major assumption in the model is that the input data accurately represent changes over the past 100 Ma in ocean pH, atmospheric pCO₂, carbonate saturation state, global temperature, and seafloor carbonate precipitation fluxes. The model fits determined in this study are only as good as the proxy data that the authors use for each of these parameters. A reader of this paper who is not familiar with the subject might be excused for thinking that we know each of these exactly as shown in the figures (e.g., Figs. 3,4,5). Yet needless to say, in each case there are robust ongoing and unresolved debates about proxy fidelity - far too many and too contentious to detail in the scope of this review. In their submission, the authors only briefly describe (almost entirely in the supplement) the assumptions that they have made in generating their records of past change, even though these fundamentally determine the model results. Their description is far too brief in my view, and there is far too little acknowledgement of the related shortcomings in the main text. The authors never evaluate (or even seem to really acknowledge) our highly imperfect knowledge of each parameter, and how this uncertainty would affect their main conclusions. Their conclusions may be robust to assumptions about "outgassing, modern fluxes, and the kinetics of seafloor weathering", as they say in the Abstract and again later in the paper, but they cannot be robust to all assumptions that are used in the inversion. For example, the authors argue that the model solutions displayed in Fig. 2 do not accurately capture the history of atmospheric pCO₂, but this

argument depends on the very high pCO₂ proxy data from the mid-Cretaceous. I think few who know the subject would doubt that pCO₂ was higher in the 90 Myrs ago compared to today, but are existing proxy data for that time sufficiently robust to be able to depend on them for the kind of quantitative analysis presented by the authors? I am not fully convinced that they are. The same could be said of temperature, pH, etc., and when you start to compound these together, I wonder how well constrained the authors' results actually are. At least I would like to see some consideration of these concerns in the paper. This is all the more the case for the seafloor weathering flux, which is a keystone of the authors' analysis, and which I would argue is relatively poorly constrained.

To some extent, I guess the authors are trying to get around questions about the details of each record focusing on the first-order variability, e.g., assuming linear changes with time, and not worrying about fitting second-order features of each curve such as warming in the Eocene, etc. That seems sensible enough, given their goals, but it does not change the reliance on limited and highly imperfect proxy data, and the total lack of acknowledgment of these imperfections. The authors' conclusions may in fact be correct, for example about a higher climate sensitivity than previously supposed, a weaker terrestrial weathering feedback, etc., but personally I think their analysis is maybe a little ahead of its time, in the sense that we are still (as a geoscience community) trying to understand each of the proxies used to infer past changes in temperature, pCO₂, etc. This concern does not take away from the value of the modeling approach itself, or its novelty, but it does lead me to question some of the bold and decisive statements made in the authors' conclusions.

To some extent, the same issues might be raised for some of the parameterizations used by the authors in setting up their model. Many of these - for example the dependence of terrestrial or seafloor weathering on temperature - are inevitably approximations. They follow a long traditional in modeling global biogeochemical cycles, of reducing complex geochemical processes like mineral dissolution into a relatively simple functional dependence. This is of course required to even begin undertaking such a global modeling task, and it's not necessarily a fatal flaw. Many past modeling efforts, for example in Berner's GEOCARB papers, have been fairly qualitative in terms of how the predictions are interpreted. Admittedly some papers have been quantitative, but I think one danger in applying a quantitative inversion as done in this case is that structural uncertainties inherent in the simplified parameterizations may lead to more confidence than is really justified in the outcome. That is to say, it seems easy to look at the solutions presented in this paper and think that we have a really good constraint on past changes in weatherability, for example, or any other shaded row in Table 1. But I am not sure that holds up to reality when considering that we don't have a particularly good idea of why seafloor weathering changed over the past 100 Ma (i.e., whether the authors' Equation 12 actually does a good job of capturing seafloor weathering global fluxes), much less how much it changed (i.e., whether the constraint used for fitting the model is accurate).

As an example of why these concerns may be important to the authors' conclusions, consider the sensitivity of weathering to climate. One of the main conclusions in this paper is that weathering is less sensitive to climate than assumed in many prior studies. Yet it seems to me, at least intuitively and without time to delve into actually replicating their model in detail, that the model-calculated climate- should depend on the magnitude of change in seafloor weathering. The authors have basically imposed a substantial change in seafloor weathering (by getting the data to fit to their estimate of seafloor weathering flux at 100 Ma), and this means they have also imposed a linear decrease in the rate of CO₂ removal. So any response of the terrestrial weathering system to changes in forcing will be muted, relative to what would be the case if seafloor weathering had not decreased by the same magnitude. In other words, by imposing a decrease in CO₂ consumption seafloor weathering, the authors are effectively imposing a limit on the climate sensitivity of terrestrial weathering. Maybe I am mistaken in this intuitive logic, but it would be good to at least have these links spelled out in the paper, so that it's easier to assess how the different conclusions depend on some of the basic underlying assumptions.

I want to be clear - I think this is an interesting paper that will add considerably to the ongoing debates about the geologic carbon cycle. I just think that there needs to be more clarity in the paper about the inherent assumptions, and a bit more of a sense of perspective about the uncertainties - not the quantitative uncertainties associated with the Bayesian inversion, which give the sense of great confidence in the outcomes, e.g., from looking at Figure 6 - but the fundamental underlying uncertainties associated with proxy fidelity and model structure, which may mean that this confidence is (to some extent or another) somewhat misguided.

Beyond these general points, I have a few specific comments on the text:

Line 28 - The carbonate-silicate weathering cycle itself was presumably "devised" by nature, or Gaia, or whoever/whatever you think made the universe with its laws of chemistry and physics - not by Walker, Kasting, and Hayes, as implied by this sentence. Arguably, Walker et al. were among the first to describe this cycle quantitatively in the scientific literature, though even then I think there is a case to be made that earlier research pioneered these ideas. Walker et al. (at a similar time to the Berner crowd) did lead in terms of the ideas about a weathering feedback.

Lines 38-39: Personally I don't see any reason why weathering fluxes to the oceans, as recorded in Be-10, should have changed over this time, just because climate cooled - especially since weathering should balance degassing, as stated earlier in the Introduction. So this sentence does not make a lot of sense to me.

Lines 88-89: It would be useful to describe how these fits were done, especially since the authors end up concluding that their model implies a much higher value of μ compared to the Brady and Gislason fits (0.75 vs. 0.23-0.32). Why is the new value so much higher? Did Brady and Gislason vastly underestimate the effective activation energy? This difference seems to merit some discussion, perhaps around lines 309-311.

Table 1: What is the "modern seafloor diss. relative precip."?

Equation 8: Is carbonate weathering flux really dependent on a weatherability factor coupled to an exponential T-dependence? I expect most surface waters draining carbonate rocks to be near saturation, such that the main control would be $p\text{CO}_2$ and runoff, not temperature.

Seafloor weathering parameterization: Most of the experimental studies cited in this section and used to justify the functional relationship between seafloor weathering and climate were conducted under far from equilibrium conditions. Are those results appropriate for seafloor pore waters? It would surprise me if pore fluids were not much closer to equilibrium, for example given their long residence times. And if that is the case, how well do we actually know the appropriate rate laws? See my comment about the uncertainties in this parameterization.

Equation 12: Presumably F_{diss} is proportional to F_{out} because outgassing is assumed to be directly proportional to crustal production rates, which in turn is assumed to regulate the rate of seafloor weathering? It would help to have this logic spelled out a bit more than as currently done in defining β . Assuming I have this right, I agree that this concept makes sense intuitively, but is there actually good evidence that seafloor carbonation rate depends on seafloor production rate? If so, it would help to provide this justification explicitly.

Figure 6: As far as I can tell (perhaps I am reading this figure incorrectly?), the parameter space between weatherability and the temperature dependence of weathering is not well constrained by the inversion, i.e., there is a valley that heads towards lower values of T_e , similar to those assumed previously, if changes in weatherability were large. Given that this parameter space seems poorly defined by the model, why do the authors rule out the possibility of a higher effective activation energy for weathering (more along the lines of the values assumed in prior

work) and a large change in weatherability? A brief additional discussion would be helpful, e.g., around lines 324-328.

Figures 7 and 8: I think these figures should have cited references in addition to the "code" for each curve (Fig. 7) and/or GCM data point (Fig. 8)? The references seem to be predominantly in the Supplement but presumably should instead be included in the main text, with clear, numbered citations in these figures.

Reviewer #3 (Remarks to the Author):

review of "The global carbon cycle since the mid Cretaceous: implications for climate sensitivity and continental versus seafloor weathering" by Krissansen-Totton and Catling

This contribution aims at showing that the weathering of seafloor basalt weathering is much more sensitive to climate than continental silicate weathering. This has implications on how the carbon cycle operates over the last 100 Myr, and on the Earth system sensitivity to a CO₂ doubling.

Adding one flux in the deep time carbon cycle, the weathering of seafloor basalts (SFW), is an hypothesis that has been tested for the first time in the early nineties (Walker 1990 in P3; François and Walker, 1992). The objective was to decouple the solid Earth degassing and the continental silicate weathering to allow the Sr isotopic composition of seawater to fluctuate. The original models were parametric and, as noted by the authors of the present contribution, they were wrong.

The present paper explore again the role of SFW on the carbon cycle over roughly the last 100 Ma. Similarly to the model developed previously by Le Hir to simulate the CO₂ pumping by SFW during snowball events, they assume that SFW depends on temperature following an Arrhenius type of law. The authors then build a simple model describing the carbon cycle. This model is first run in a direct way. Then it is run following a bayesian inverse method to constrain the values of the 16 main parameters of the model. This inversion is the innovative part of this contribution, leading to interesting results such as the quantification of the required rise in continental weatherability from the Cretaceous to the present day.

However, I have several questions:

- 1) Everything relies on the observation that SFW was more intense during the mid-Cretaceous climatic optimum than at present day. This is taken from the study of Coogan and Dosso. But this study gives only one point in the past. What about the evolution over the late Cretaceous and the whole Cenozoic ? The authors reconstruct a long term decrease from the high Cretaceous value towards the present day, without supporting data for this scenario. There is a growing evidence that the late Cretaceous was much cooler than the Cenomanian-Turonian episode (see for instance Puçeat et al 2007 in Geology). The high mid Cretaceous values might correspond to a short-lived peak, and the inversion method used here would produce drastically different results.
- 2) I would not say that the calculated parameter values are "robust" (this appear at several places in the paper) only because a statistical method has been applied. The results heavily depends on the way equations are written. If the equations are mechanistic, I would agree with the robustness argument. But it is not the case here. There are many parameterizations in the model, some of them new: link between CO₂ and surface T, between surface T and deep T, dependence on pH of the SFW, description of the carbonate deposition in the pelagic environment, continental weathering, the link between CO₂ and DIC... The results are necessarily dependent on the way equation are written.
- 3) I feel frankly uncomfortable with the criticism regarding the model of Le Hir 2008. Those critics are wrong and this must be corrected. First, the Le Hir model accounts for the dissolution of the seafloor in acidic environment and in basic environment (it is a sum of terms). Line 630, it is stated that the Le Hir model is in contradiction with the experimental results showing a U-shaped dependence of weathering on pH. But the model of Le Hir uses laboratory kinetic laws fitting the U-shaped curve, so it is fully consistent with experimental results. In Le Hir 08, last line, it is stated that weathering is minimal at pH 8.2. Conversely, the present model does not reproduce the U-shaped curve since it is a function of H to the power gamma. So the question becomes: why do the authors develop a model less efficient than the Le Hir one ? Furthermore, I can't find in Le Hir any discussion about the dominance of basaltic glass dissolution. Where does the discussion from line 625 to line 630 come from ?
- 4) It is also stated on line 104-106 that the Le Hir model ignores the effect of temperature. Once again, this is not true. The Le Hir model includes an Arrhenius law to account for temperature.

But, and this is an important point, Le Hir sets the temperature of fluids percolating the oceanic crust to a constant 40°C, while it is set at the deep ocean temperature plus 9K in the present contribution. Alt and Teagle (99) state that the temperature of water-rock interaction fluctuates from 0 to 60°C, depending on the depth of percolation. Because percolation is not modelled in Le Hir (2008), they choose to fix the temperature to a constant value, independently from the deep ocean temperature. Conversely, in the present contribution, the authors assume that the interaction occurs at the deep seawater temperature plus 9K. This 9K value is taken from Coogan and Dosso (2015). Curiously, while all the main parameters of the model are being tested, this 9K value is assumed to be a well-constrained constant. Looking at the Coogan and Dosso paper, this constant can fluctuate from 1 to 20°C. Using the highest value displayed in table 1 for E_a (107 kJ/mol/K) to maximize the effect, I plotted below the factor of increase in SFW when temperature of the deep ocean rises from 3 to 12°C, assuming a correction of 1K, 9K, and 20K. I do not know if this matters or not, but it should be checked.

- 5) line 511-515: the authors may say that they are neglecting the role of organic carbon, but they cannot state that there is no secular change in organic carbon burial fraction over the last 100 My. Check Godderis and François, GRL, 1996; France-Lanord and Derry, Nature, 1997; Galy et al., Nature, 2007. Organic carbon disequilibria are a primary driver of the Cenozoic carbon cycle. Also have a look at Katz et al., 2005, Marine Geology
- 6) The first order controlling factor of continental weathering is runoff, and runoff heavily depends on paleogeographies (Otto-Bliesner, 1995, JGR; Gibbs et al., 1999, P3; Godderis et al., 2014, ESR). How is this dependency included in the present modeling ?
- 7) line 177: the climate sensitivity evolves through time with the paleogeography (Godderis et al, 2014, ESR). This important effect is neglected here.

In summary, this is an interesting study, but I think that the above list of questions precludes its publication in Nature Communication. Overall, it remains rather technical and I think it should be submitted to a more specialized journal.

Reviewer #4 (Remarks to the Author):

This paper presents a statistically thorough assessment of the factors in the carbonate-silicate cycle that have stabilized the carbon cycle and presumably climate over the last 150 million years. This is a subject of great interest and significant prior work, but the current study is perhaps the most conclusive because of the way Bayesian statistics is applied to assess uncertainties in the parameters and forcing factors. The paper comes to a number of convincing if contrary-to-current-thinking conclusions, including that climate sensitivity of CO₂ consumption during silicate weathering is much lower than previously assumed, that weatherability was lower in the Cretaceous than today (actually this is convention but the treatment here more exhaustive), climate sensitivity was above the IPCC range, and that seafloor weathering, while important (the vanguard thinking) was still dominated by continental weathering during this time interval. These are all important and interesting conclusions, and I recommend publication with modest revision:

1) I'll upload a paper we wrote in 1997 that lays out the concept of weatherability as providing a resolution to the Bernver vs. Raymo arguments about weathering rates and tectonics. In that we calculate a weatherability increase for the Cenozoic that is not unlike that presented here. We also went a bit further to consider imbalances in the organic C cycle, shelf-pelagic carbonate partitioning, etc., implications for Sr isotopes. I'd think the authors might want to cite this on line 42 but also consider comparing our results to theirs later on in the manuscript.

2) Line 38 and on: the fact that global weathering rates haven't changed much in the past 12 Ma despite cooling is not inconsistent with the overall carbonate silicate "thermostat" idea: indeed, in the absence of changes in volcanism, silicate weathering rates are predicted to stay constant despite cooling (indicating that the cooling is ultimately being driven by other factors like weatherability).

3) Line 41: I wouldn't lump runoff into "weatherability" since runoff is intimately tied to climate and CO₂; weatherability factors should be external to the climate system (at least in terms of direct relationships).

4) Line 58: word missing "Radiometric ... confirms"

5) Since the authors are being thorough and precise elsewhere, they need to revise the statement concerning alkalinity (line 63 and on) which incorrectly states that carbonate alkalinity (less water dissociation) must balance the conservative ion charge imbalance: much more important than water dissociation are borate, silicate, phosphate, etc. all the weak acid anions in seawater.

6) The inability to disentangle variations in outgassing from variations in weatherability has a history to it. I think first of the Kasting paper from 1984 (AJS) that showed that the BLAG model prediction of pCO₂ all boiled down to the quotient of outgassing rate to land area (the one "weatherability" factor considered in BLAG). The authors portray this as perhaps more under constrained than it really is: Rowley (1997) has assessed likelihood of spreading rate variations through time, for example.

7) Can the authors comment explicitly on whether their results argue for transport vs reaction limitation for the present and past worlds, a la Stallard and Edmond 1983?

8) We did the basalt exposure through time calculation several years before Mills et al.; see Bluth and Kump (1991; AJS 291:284-308; see also 1994: GCA 58:2341-2359 and for GCM-based weathering calculations last 250 Ma, Gibbs et al. 1999, AJS 299: 611-651; applies to discussion line 399 and on concerning ref 34).

Line 386: no hyphen for ..ly modifiers.

Line 406: What exactly makes a pCO₂ of 4000 ppm unreasonable? Most of the proxies we use saturate below that level. I don't think we can rule them out.

Response to Reviewer 1

Paper: The global carbon cycle since the mid Cretaceous: Implications for climate sensitivity and continental versus seafloor weathering

Authors: Krissansen-Totton and Catling

Reviewer: Dorian S. Abbot

Date: July 1, 2016

Overview: The authors build a new carbon cycle model and perform a Bayesian statistical analysis using it that allows them to infer parameter values of their model based on other people's observations. Their most interesting conclusions are that continental weathering responds much more weakly to temperature than we had previously realized, and that seafloor weathering does seem to respond to the surface climate, allowing a negative climate-seafloor weathering feedback. The paper is clear and the arguments are well-presented. The work is impressive and brings together a huge amount of data to make relevant and exciting inferences. I consider this work significantly more sophisticated than what is typically seen in paleoclimate/paleoceanography papers. I think this paper deserves quick publication and will have a large impact on the field. I give a few comments for improvement below.

Comments:

1. Climate Sensitivity: I am not convinced by the climate sensitivity inference for a number of reasons. The first is just that we do not have any information about other greenhouse gasses such as methane (and we have reason to suspect that they might be higher in a hot-house climate). Neglecting these leads to a big upward bias in the inferred climate sensitivity.

Also, I was not able to find a description of the CO₂ proxy data. Maybe I just missed it, or maybe it could be in a more prominent place. Anyway, I do not see error bars on the data points from the Cenozoic when these are plotted in the B panels of the relevant figures. What are you assuming for the errors when you use these in the Bayesian analysis? Also, are these from leaf stomata? That method is notoriously unreliable, especially when trying to infer high CO₂ values because the method saturates. If the CO₂ really were higher during the Eocene than you think, the climate sensitivity you infer would be lower than it should be.

We have revisited all the proxy data in response to comments by Reviewer 2, but we have paid special attention to CO₂ proxies in response to the comments above. The CO₂ data and our binning methodology are now fully described in Section S4. We now plot error bars for Cenozoic data. The Cenozoic data is an extensive compilation of many different proxy techniques (stomatal, paleosol, phytoplankton, liverwort, and boron-based estimates) and thus represents a consensus reconstruction. Since we binned this data by taking the range between the lowest and highest pCO₂ estimates, we are not systematically biased toward higher or lower climate sensitivities. We have also revisited Cretaceous pCO₂ data. In the original manuscript we were only using paleosol data, but in the revised manuscript this is complimented by stomatal data and fossil bryophytes. Stomatal methods do tend to produce lower pCO₂ estimates than paleosol methods, but our reading of the literature suggests this discrepancy is unresolved (see the new discussion in Section S4), and so the most conservative approach is to include all data in our inversion and assume large uncertainties. However, even if Cretaceous stomatal data are omitted – such as in the original manuscript – the inferred climate sensitivity is still large.

Finally, I am not satisfied with the claim that changes in insolation roughly balance changes in radiative forcing due to continental configuration. There are two major issues with this: (1) the two effects have different patterns in time and there's way they perfectly canceled over the entire 100 Myr period considered and (2) the radiative forcing due to changing the continental configuration is going to be model dependent, mainly due to cloud effects (just because one model says it cancels with insolation does not mean this is true when you "marginalize" over reasonable models). For example, consider the Eocene. If we assume a planetary albedo of 0.3, then the solar forcing was something about 1-1.5 W m⁻² lower during the Eocene. If we assume a climate sensitivity of 0.8 KW⁻¹ m², we get about 1 K of cooling due to the lower solar forcing. In CCSM3, going to Eocene configuration causes 5 K of warming (Caballero and Huber, 2013), although it's a little unclear how much of that to count as forcing and how much as Earth system sensitivity.

Anyway, if you think the Eocene CO₂ is low because of (questionable) leaf stomata proxy data, you leave out 4 K of net warming due to changes in forcing, and you do not consider forcing by other greenhouse gasses, you are going to infer a climate sensitivity that is way too high!

The bottom line is that I would try a little harder on the CO₂ proxy error bars, I would try to include variations in solar forcing and forcing due to changes in continental configuration, and I would play down the claim to have inferred climate sensitivity since we have no information on other greenhouse gasses (or maybe call the inference you make an upper limit).

We have modified our climate model in several ways to accommodate these suggestions. Our climate model now includes changes in solar luminosity since 100 Ma, as described in the new Section S2. We have accounted for luminosity changes in a general way by tying them to our unknown climate sensitivity. Additionally, we now account for paleogeography changes in our climate model. The reviewer is correct to highlight the fact that the radiative forcing from changes in continental configuration is model dependent, and so assuming that paleogeography cancels luminosity is not necessarily a robust approach. In the revised manuscript we instead assume a range 0-5 K from paleogeography forcings since 100 Ma. This range is consistent from several different GCM studies (Barron et al., 1993; Sloan & Rea, 1995; Heinemann et al., 2009; Dunkley Jones et al., 2010 as cited in Royer et al. 2012). The Caballero and Huber study that the reviewer cites provides the upper limit on this range, and is now cited in the main text.

Regarding methane, we think it's unlikely that the omission of methane will dramatically overestimate Earth system climate sensitivity. Coupled GCM Earth-system models suggest that other greenhouse gases may have contributed significantly (2-3 K) to Cretaceous and Eocene warmth (e.g. Beerling et al. 2011, PNAS). However, this increase in GHG abundances is a feedback response to warmer temperatures, namely vegetation changes and subsequent atmospheric chemistry changes. Methane *feedbacks* are already implicitly captured in our climate equation. In principle, the omission of methane *forcings* could change our results, but the changes in flux would have to be substantial, and we are not aware of any literature arguing for large changes in methane fluxes due to changes in boundary conditions alone. In Beerling et al. (2011), changes in boundary conditions between the Early Eocene and Late Cretaceous account for only 0.5 K warming from other greenhouse gases. This would have a minor effect on our climate sensitivity posterior distribution. Furthermore, the changes in vegetation in Beerling et al. (2011) are broadly consistent with paleobotanical reconstructions (Shellito and Sloan, 2006),

suggesting there is no need to invoke large increases in methane fluxes driven by boundary condition changes.

With that said, we agree that high Cretaceous methane due to changing boundary conditions cannot be completely ruled out, and so we added a paragraph to the discussion explaining this caveat, summarizing the discussion above, and pointing out the potential implications for climate sensitivity. Additionally, in the conclusions section we have added a line that clearly states our assumptions about methane in the model.

2. Importance of weathering: The work you have done gives you the chance to do some cool calculations of the importance of weathering feedbacks. For example, you could run the forward model with all weathering feedbacks turned off and see how the climate evolves. You could also switch off either the continental or seafloor weathering. This would be a nice way to show which one is more important. I think this would help make the paper accessible to the paleoclimate crowd, who may get lost in the Bayesian analysis.

These simple forward model tests are less informative than you might expect and/or potentially easy to misinterpret:

Case 1: No weathering feedbacks

If all weathering feedbacks are switched off then pCO₂ increases dramatically to ~1 bar at 100 Ma with surface temperatures around 70°C (in fact surface temperatures should be much higher because our logarithmic climate function is a poor approximation to radiative transfer models at high pCO₂). This result is essentially identical to those of previous studies such as Berner and Caldeira (1997) who showed a negative feedback is needed to balance the carbon cycle and avoid unphysical scenarios. We believe this result is widely understood and does not require reproduction in this study.

Case 2: No Continental weathering feedback ($\alpha=0$, $T_e=\text{infinity}$)

If the continental weathering feedback is switched off then the model $p\text{CO}_2$ is too high relative to proxies (100 Ma median $p\text{CO}_2 \sim 0.01$ bar). This suggests that seafloor weathering alone probably cannot explain the last 100 Ma. However, this should not be misinterpreted as implying that the seafloor weathering feedback is always weak. Rather, seafloor weathering is unable to keep atmospheric $p\text{CO}_2$ low because the initial (modern) flux is small relative to the initial continental flux. If the modern seafloor weathering flux is set equal to the modern continental weathering flux, then seafloor weathering feedbacks are easily able to maintain physically reasonable $p\text{CO}_2$ levels:

Case 3: No seafloor weathering feedback ($\gamma=0.0$, $E_{bas}=0.0$, $\beta=0.0$):

Case 3: If seafloor weathering feedback is switched off then the model output does not look too different to Fig. 4 in the main text, although the distributions are shifted somewhat. This result could be misinterpreted as implying that seafloor weathering is unimportant. However, in practice it is difficult to determine the importance of seafloor weathering giving the magnitude of the uncertainties in other variables. For example if the continental weathering parameters are assumed to take on values near the ‘weaker’ end of their distributions, seafloor weathering is switched off, and the outgassing parameter is slightly restricted, then there is complete misfit with proxies:

This shows that seafloor weathering could be an important sink, but we can't really say because the uncertainties in other parameters are so large. Ultimately it's hard to say anything definitive with this kind of forward modeling because there are so many degeneracies. This is why the manuscript focuses on the inverse analysis to determine what can (and cannot) be definitively constrained. Indeed, in the inverse analysis we find that the seafloor weathering parameters (γ, E_{bas}, β) are not particularly well constrained, reflecting the forward model results above.

We believe that including these cases in the main text would not improve clarity and might instead result in our work being misinterpreted.

3. Details: Figure 6 is supposed to contain posterior distributions of well-constrained variables. α , T_e , and $1+V$ do not seem very well-constrained, in that the posterior does not seem to taper to low values at their extremes. Would it make sense to try larger ranges in these variables so that you sample values that the data can exclude?

There are good reasons to exclude larger ranges with the prior. For example, empirical results constrain α , and estimates of Cretaceous outgassing (Fig. 7) from a variety of different techniques span a similar range. Expanding the prior beyond these plausible ranges allows solutions that conflict with current knowledge.

As a minor point, we are careful not to claim that Fig. 6 only shows 'well-constrained' variables in the text since clearly it shows a mixture of well-constrained and poorly constrained variables. It is simply a selection of interesting variables.

Why not present the temperature dependence of seafloor and continental weathering using the same units so they can be compared more easily?

We do convert the continental weathering e-folding temperature to an activation energy so that it can be compared to the activation energy of seafloor weathering. It is stated in both the results section and the conclusions that the activation energy for continental weathering is $20+10/-5$ kJ/mol, which has the same units as the 53-97 kJ/mol 1σ range we derive for seafloor weathering. However, we intentionally do not emphasize this comparison because it is misleading. Continental weathering responds directly to CO₂ whereas seafloor weathering does not, and so any comparison of the response of continental and seafloor weathering to a change in climate needs to account for both factors. And of course the silicate weathering response is also a function of climate sensitivity which adds uncertainty to the comparison.

You note that continental weathering dominates over seafloor weathering, but it seems that the changes in these variables are more important than their total magnitudes if we care about a negative weathering feedback. It's a little hard to tell from Figure 5E,F which is changing more because of the difference in the scales, but this would be worth discussing. In the revised manuscript we have included a subfigure that directly compares the (absolute) changes in seafloor weathering and continental weathering (Fig. 5 – the relative change distributions have been combined into a single figure to save space). The absolute change in continental weathering is probably greater than the absolute change in seafloor weathering, but the reverse could be true. The significance of this result is explored in a new paragraph in the revised discussion.

References

Caballero, R., and M. Huber (2013), State-dependent climate sensitivity in past warm climates and its implications for future climate projections, *Proceedings of the National Academy of Sciences*, 110(35), 14,162–14,167.

Response to Reviewer 2

Reviewer #2 (Remarks to the Author):

In this paper, the authors apply a new model to evaluate the mechanisms operating to steer changes in the global carbon cycle over the past 100 million years. One significant addition relative to prior studies on this topic comes in the way that the authors treat seafloor weathering. Though not totally ignored in previous work, this component of the carbon cycle has not had the treatment that it deserves (or needs), so this aspect of the authors' contribution is a particularly welcome addition. The authors conclude that seafloor weathering has played a seminal role in charting the course of C cycle evolution, more than previously thought, and that the responsiveness of terrestrial weathering to changes in climate (the "negative weathering feedback" that stabilizes long-term climate) has been over-estimated in past work. The authors' results also suggest that global "weatherability" has changed over the past 100 Ma, confirming inferences made in a number of prior studies.

Nonetheless, as far as I can tell this study can only weakly constrain the magnitude of change in weatherability and cannot rule out equally or more important changes in outgassing, in terms of contributing to the overall cooling trend since the Cretaceous. In other words, the authors are not able in this study alone to answer the "uplift vs. degassing" debate, for what that duality is worth. In any case, the authors here are more concerned about understanding how feedbacks have operated to maintain a stable climate system. Their conclusions in these regards are without a doubt interesting and novel, i.e., about relative roles of seafloor vs. terrestrial weathering, and about the strength of climate sensitivity and weathering feedbacks. Overall, at least from my perspective, these are important questions, so this promises to be a widely read study on a topic of significant interest in the geosciences, in an area where a number of questions remain to be fully resolved. And in addition to the provocative interpretations, I am confident that the paper provides some new modeling directions that may help take the field forward.

On the other hand, in my view a substantial weakness of this paper is that the authors arrive at a number of conclusive statements that I am not sure are fully supported by their analysis. Or at least I think the authors are more definitive in the way that they present their conclusions than seems really warranted to me. The authors use a Bayesian inversion to arrive at a set of parameters that they argue best describe the evolution of the C cycle. Because their model arrives at a best-fit solution, they conclude that the best-fit parameters must describe the actual system behavior. While I agree that this kind of inverse approach has a lot to offer in addressing the problem at hand, I think the paper in its current form is at risk of being overly conclusive, principally because the model itself depends on poorly known inputs.

A major assumption in the model is that the input data accurately represent changes over the past 100 Ma in ocean pH, atmospheric pCO₂, carbonate saturation state, global temperature, and seafloor carbonate precipitation fluxes. The model fits determined in this study are only as good as the proxy data that the authors use for each of these parameters. A reader of this paper who is not familiar with the subject might be excused for thinking that we know each of these exactly as shown in the figures (e.g., Figs. 3,4,5). Yet needless to say, in each case there are robust ongoing and unresolved debates about proxy fidelity - far too many and too contentious to detail in the scope of this review. In their submission, the authors only briefly describe (almost entirely in the supplement) the assumptions that they have made in generating their records of past change, even though these fundamentally determine the model results. Their description is far too brief in my view, and there is far too little acknowledgement of the related shortcomings in the main text. The authors never evaluate (or even seem to really acknowledge) our highly imperfect knowledge of each parameter, and how this uncertainty would affect their main conclusions. Their conclusions may be robust to assumptions about "outgassing, modern fluxes, and the kinetics of seafloor weathering", as they say in the Abstract and again later in the paper, but they cannot be robust to all assumptions that are used in the inversion. For example, the authors argue that the model solutions displayed in Fig. 2 do not accurately capture the history of atmospheric

pCO₂, but this argument depends on the very high pCO₂ proxy data from the mid-Cretaceous. I think few who know the subject would doubt that pCO₂ was higher in the 90 Myrs ago compared to today, but are existing proxy data for that time sufficiently robust to be able to depend on them for the kind of quantitative analysis presented by the authors? I am not fully convinced that they are. The same could be said of temperature, pH, etc., and when you start to compound these together, I wonder how well constrained the authors' results actually are. At least I would like to see some consideration of these concerns in the paper. This is all the more the case for the seafloor weathering flux, which is a keystone of the authors' analysis, and which I would argue is relatively poorly constrained.

1) In the revised manuscript we have greatly expanded our discussion of proxy data and their limitations. The new Section S4 describes the proxy data and limitations for the five variables we considered. The revised discussion section contains a paragraph discussing how our results are contingent on uncertain proxy data, the wording of the conclusions has been modified to remind the reader that our results are dependent on uncertain proxies, and there is a new section in the results section that explores the sensitivity of our results to different proxies.

At the reviewer's suggestion we have thoroughly reexamined our data selection to ensure they represent the diversity of opinions in the literature and not merely one side of a contentious debate. For example, the reviewer highlighted that the original manuscript assumed high pCO₂ in the mid Cretaceous. In the revised manuscript we have expanded our CO₂ dataset to include both high and low mid Cretaceous estimates from alternative proxies (the different proxies are discussed in Section S4). The interpretation of the goodness-of-fit in the forward modeling has been reevaluated accordingly.

Additionally, we now use a very conservative method to bin proxy data, thereby ensuring that inverse modeling is not driven by unwarranted precision in proxy estimates. For every 10 Ma year interval, our 'best-estimate' envelope for any given geochemical proxy is taken to be the range of data plus uncertainties in that interval (for example see Fig. S8, S9, S10 and S11). If anything, this approach will overestimate the uncertainties in proxies because we set error bars of binned data to encompass the entire range of literature estimates.

Of course, the possibility exists that some proxy estimates from the literature are simply incorrect. This would change our results to varying degrees depending on the proxy. In the revised text we have added a new section to the results section that explores the sensitivity of our results to different proxies by re-running the inverse analysis omitting each variable one-by-one (and sometimes omitting multiple variables from the analysis). The significance of these calculations is highlighted in a new paragraph in the revised discussion. In short we find that our conclusions on weak temperature-dependence of continental weathering and weatherability changes are robust. However, climate sensitivity results are sensitive to the accuracy of pCO₂ and temperature proxies we have adopted.

A key goal of this study was to incorporate seafloor weathering into a carbon cycle in a physically plausible way. However, it is not true that the seafloor weathering constraints are a 'keystone' of our analysis – in fact our main conclusions are largely not dependent on the seafloor weathering sink we have assumed (see response to a similar comment below for further explanation).

To some extent, I guess the authors are trying to get around questions about the details of each record focusing on the first-order variability, e.g., assuming linear changes with time, and not worrying about fitting second-order features of each curve such as warming in the Eocene, etc. That seems sensible enough, given their goals, but it does not change the reliance on limited and highly imperfect proxy data, and the total lack of acknowledgment of these imperfections. The authors' conclusions may in fact be correct, for example about a higher climate sensitivity than previously supposed, a weaker terrestrial weathering feedback, etc., but personally I think their analysis is maybe a little ahead of its time, in the sense that we are still (as a geoscience community) trying to understand each of the proxies used to infer past changes in temperature, pCO₂, etc. This

concern does not take away from the value of the modeling approach itself, or its novelty, but it does lead me to question some of the bold and decisive statements made in the authors' conclusions.

2) One result from the sensitivity analysis (with revised proxies) is that our climate sensitivity is strongly dependent on temperature proxies, which are uncertain. We have added this caveat to our conclusions to soften our statements on climate sensitivity. Nonetheless, we believe strong statements on the temperature-dependence of continental weathering and the change in weatherability since 100 Ma are warranted for the reasons outlined above. However, we have prefaced all our conclusions with a reminder that our results are contingent on the reliability of current proxies.

To some extent, the same issues might be raised for some of the parameterizations used by the authors in setting up their model. Many of these - for example the dependence of terrestrial or seafloor weathering on temperature - are inevitably approximations. They follow a long tradition in modeling global biogeochemical cycles, of reducing complex geochemical processes like mineral dissolution into a relatively simple functional dependence. This is of course required to even begin undertaking such a global modeling task, and it's not necessarily a fatal flaw. Many past modeling efforts, for example in Berner's GEOCARB papers, have been fairly qualitative in terms of how the predictions are interpreted. Admittedly some papers have been quantitative, but I think one danger in applying a quantitative inversion as done in this case is that structural uncertainties inherent in the simplified parameterizations may lead to more confidence than is really justified in the outcome. That is to say, it seems easy to look at the solutions presented in this paper and think that we have a really good constraint on past changes in weatherability, for example, or any other shaded row in Table 1. But I am not sure that holds up to reality when considering that we don't have a particularly good idea of why seafloor weathering changed over the past 100 Ma (i.e., whether the authors' Equation 12 actually does a good job of capturing seafloor weathering global fluxes), much less how much it changed (i.e., whether the constraint used for fitting the model is accurate).

3) This is an important caveat that is now highlighted in the revised discussion. However, we don't think this problem is as severe as the reviewer suggests. Many of the functional forms in our model are not arbitrary parameterizations but are based on well-understood physics and/or empirical constraints. Here are a few examples:

- The logarithmic dependence of surface temperature on $p\text{CO}_2$ is a good approximation to radiative transfer model outputs.
- The exponential temperature dependence of weathering is a consequence of molecular kinetics. The exponential-form Arrhenius equation is derivable from Transition State Theory (e.g. Rimstidt, 2014, *Geochemical Rate Models*, p. 84).
- The linear relationship between deep and surface ocean temperatures is based on a large amount of data, namely paleoclimate proxies from many different periods in Earth's history and an ensemble of GCM outputs which capture the physics of ocean circulation and heat transfer.
- The function relating carbonate precipitation to saturation is based on a number of empirical and experimental studies.

That being said, we also use parameterizations that are less well grounded in physics or empirical results out of necessity. But in these cases we tried to define the parameterizations in the most general way possible such as to ensure that our results were robust (or at least as robust as possible) to uncertainties in the functional representation of biogeochemical processes. Specifically, we use power laws with unknown exponents to capture these uncertain processes. Here are a few examples:

- In the seafloor weathering equation (equation 12) that the reviewer highlights, the uncertain relationship between outgassing, crustal production, and basalt dissolution is captured with a power law with an exponent varying all the way from 0 (changes in both crustal production and outgassing have had no impact on basalt dissolution on the seafloor) to 1 (all the outgassing changes in the last

100 Ma can be attributed to changes in crustal production, and there is a one-to-one relationship between crustal production and seafloor weathering).

- The relationship between atmospheric $p\text{CO}_2$ and continental weathering is modeled as an unknown power law. The bounds for the exponent (0.2-0.5) are based on experimental results with a range of minerals and ensemble of exchange models linking atmospheric CO_2 to soil CO_2 .
- The relationship between hydrogen activity and basalt dissolution is also a power law. The bounds of the exponent vary from 0 (no pH dependence, as supported by some experimental studies), to 0.5 (a strong relationship and upper bound to some experimental studies).

Although we are confident in the parameterizations we have adopted, the possibility remains that our simple functions do not fully capture the geochemical processes we are attempting to represent. Some relationships may be stochastic rather than mechanistic or have functional forms that change with time. However, these limitations apply to all biogeochemical models, and so other than acknowledging these limitations in the main text, we don't think additional changes are warranted.

As an example of why these concerns may be important to the authors' conclusions, consider the sensitivity of weathering to climate. One of the main conclusions in this paper is that weathering is less sensitive to climate than assumed in many prior studies. Yet it seems to me, at least intuitively and without time to delve into actually replicating their model in detail, that the model-calculated climate- should depend on the magnitude of change in seafloor weathering. The authors have basically imposed a substantial change in seafloor weathering (by getting the data to fit to their estimate of seafloor weathering flux at 100 Ma), and this means they have also imposed a linear decrease in the rate of CO_2 removal. So any response of the terrestrial weathering system to changes in forcing will be muted, relative to what would be the case if seafloor weathering had not decreased by the same magnitude. In other words, by imposing a decrease in CO_2 consumption seafloor weathering, the authors are effectively imposing a limit on the climate sensitivity of terrestrial weathering. Maybe I am mistaken in this intuitive logic, but it would be good to at least have these links spelled out in the paper, so that it's easier to assess how the different conclusions depend on some of the basic underlying assumptions.

4) This particular example is not problematic. When we repeated the inversion but omitted the seafloor weathering sink the key conclusions about the temperature dependence of continental weathering and climate sensitivity do not change. Similarly, there is only a small shift in the retrieved value for the weatherability increase since 100 Ma (see results section revised manuscript). This result is seemingly counterintuitive because, as the reviewer points out, including seafloor weathering does impose a decreasing CO_2 sink. However, the magnitude of the seafloor weathering sink is small compared to continental silicate weathering, and so its omission has only a secondary effect on the retrieval. Seafloor weathering had only a small influence on results in the submitted manuscript and continues (in the revised version) to have little influence. Instead, weakly T-dependent continental weathering and change in continental weatherability are primarily a result of the model fit to $p\text{CO}_2$, pH, temperature, and saturation proxies. This is explained in the revised results section.

I want to be clear - I think this is an interesting paper that will add considerably to the ongoing debates about the geologic carbon cycle. I just think that there needs to be more clarity in the paper about the inherent assumptions, and a bit more of a sense of perspective about the uncertainties - not the quantitative uncertainties associated with the Bayesian inversion, which give the sense of great confidence in the outcomes, e.g., from looking at Figure 6 - but the fundamental underlying uncertainties associated with proxy fidelity and model structure, which may mean that this confidence is (to some extent or another) somewhat misguided.

5) We hope the modifications described above have clarified assumptions and highlighted uncertainties.

Beyond these general points, I have a few specific comments on the text:

Line 28 - The carbonate-silicate weathering cycle itself was presumably "devised" by nature, or Gaia, or whoever/whatever you think made the universe with its laws of chemistry and physics - not by Walker, Kasting, and Hayes, as implied by this sentence. Arguably, Walker et al. were among the first to describe this cycle quantitatively in the scientific literature, though even then I think there is a case to be made that earlier research pioneered these ideas. Walker et al. (at a similar time to the Berner crowd) did lead in terms of the ideas about a weathering feedback.

6) Wording has been changed to ensure Walker et al. are credited with the cycle's description.

Lines 38-39: Personally I don't see any reason why weathering fluxes to the oceans, as recorded in Be-10, should have changed over this time, just because climate cooled - especially since weathering should balance degassing, as stated earlier in the Introduction. So this sentence does not make a lot of sense to me.

7) This sentence has been removed from the revised manuscript.

Lines 88-89: It would be useful to describe how these fits were done, especially since the authors end up concluding that their model implies a much higher value of μ compared to the Brady and Gislason fits (0.75 vs. 0.23-0.32). Why is the new value so much higher? Did Brady and Gislason vastly underestimate the effective activation energy? This difference seems to merit some discussion, perhaps around lines 309-311.

8) There is nothing obviously wrong with the Brady and Gislason fits. Rather, estimates of the activation effective energy of basaltic dissolution vary widely, presumably due to different experimental procedures and the challenges of extrapolating laboratory rates to ~million year timescales. We have added a discussion around lines 304-309 contrasting our activation energy with that of experimental and field studies, and suggesting why they may differ.

Additionally, we have decided to remove the 0.75 vs 0.23-0.32 comparison from the revised manuscript because it is potentially misleading. Readers might conceivably adopt the $\mu=0.75$ value in their own models, which would be incorrect. The temperature dependence of seafloor weathering cannot be reduced to an overall CO_2 dependence in this way. This is because as we go back in time pCO_2 gradually increases due to declining solar luminosity, but surface temperature remains relatively constant. Converting the temperature dependence of seafloor weathering to a pCO_2 dependence and using the RCO_2^{μ} equation to predict seafloor weathering will overestimate seafloor weathering because the CO_2 increase does not reflect a change in temperature but rather the buffering of the continental silicate weathering feedback.

Table 1: What is the "modern seafloor diss. relative precip."?

9) The wording of this entry has been changed and a footnote has been added clarifying what this means.

Equation 8: Is carbonate weathering flux really dependent on a weatherability factor coupled to an exponential T-dependence? I expect most surface waters draining carbonate rocks to be near saturation, such that the main control would be pCO_2 and runoff, not temperature.

10) Most carbon cycle models that include carbonate weathering do incorporate some temperature dependence (e.g. Berner 2004; Mills et al. 2014 PNAS; Francois and Walker 1992; Kashiwagi et al. 2008). This is because - as the reviewer correctly notes - carbonate weathering is a function of runoff, and runoff is temperature dependent. Of course, the runoff-only temperature dependence is probably going to be weaker than the runoff+kinetics temperature dependence of silicate weathering, and so using the same temperature exponential for both silicates and carbonates is inaccurate. However, we allow for the possibility that

carbonates and silicates have different temperature dependencies by including an additional coefficient in our carbonate weathering expression. This is explicitly stated in the methods section: “we assume carbonate weathering has the same functional form as silicate weathering, except for an additional dimensionless multiplicative factor, C_{WF} , to allow for the possibility that carbonate weathering is subject to **different temperature-dependence**, CO_2 -dependence, and weatherability factors.” This carbonate weatherability factor has an unknown linear trend over 100 Ma, and we assume a wide range of gradients to allow for the possibility that carbonate weathering is very different to silicate weathering. To clarify this, we have modified the methods section to explain how carbonate weathering is primarily a runoff effect, and so the weatherability factor is required.

In any case we have done tests where we have run the model with no carbonate weathering temperature-dependence, and found the results to be the same. This is because the broad range of carbonate weatherability factor changes dwarfs any subtle differences in explicit temperature dependence.

Seafloor weathering parameterization: Most of the experimental studies cited in this section and used to justify the functional relationship between seafloor weathering and climate were conducted under far from equilibrium conditions. Are those results appropriate for seafloor pore waters? It would surprise me if pore fluids were not much closer to equilibrium, for example given their long residence times. And if that is the case, how well do we actually know the appropriate rate laws? See my comment about the uncertainties in this parameterization.

11) The experimental studies we cited were indeed conducted under far-from-equilibrium conditions. However, the Coogan and Dosso (2015) paper cited in the text constrains the seafloor weathering-climate relationship using an empirical fit to real-world data (oxygen isotopes and Sr isotopes are used as proxies for ocean temperature and seafloor dissolution, respectively). The temperature dependence derived from this empirical approach is broadly consistent with experimental results, and is able to reproduce the Cenozoic Sr isotope record. Additionally, we allow for a very broad range of rate laws in our analysis, with effective activation energies ranging from 40 to 110 kJ/mol – this encompasses the values reported in both empirical and experimental studies.

Equation 12: Presumably F_{diss} is proportional to F_{out} because outgassing is assumed to be directly proportional to crustal production rates, which in turn is assumed to regulate the rate of seafloor weathering? It would help to have this logic spelled out a bit more than as currently done in defining beta. Assuming I have this right, I agree that this concept makes sense intuitively, but is there actually good evidence that seafloor carbonation rate depends on seafloor production rate? If so, it would help to provide this justification explicitly.

12) Yes this is correct. We have changed the wording of the main text to clarify this point. There is no direct evidence for this relationship. To establish a correlation between crustal production and the seafloor weathering sink would require accurate knowledge of crustal production rates through time, something that is contentious (see Fig. 7). This is why we have adopted a generalized power law with an exponent that can vary from zero to one. There is uncertainty in how outgassing relates to crustal production, and there is uncertainty in how crustal production relates to the seafloor weathering sink.

Figure 6: As far as I can tell (perhaps I am reading this figure incorrectly?), the parameter space between weatherability and the temperature dependence of weathering is not well constrained by the inversion, i.e., there is a valley that heads towards lower values of T_e , similar to those assumed previously, if changes in weatherability were large. Given that this parameter space seems poorly defined by the model, why do the authors rule out the possibility of a higher effective activation energy for weathering (more along the lines of the values assumed in prior work) and a large change in weatherability? A brief additional discussion would be helpful, e.g., around lines 324-328.

13) We imposed a prior on the weatherability change whereby the minimum mid Cretaceous value was 20% the modern value. This cut-off is based on published estimates of how different variables may affect

weatherability. For instance Cretaceous continental area was 73-90% modern (see in text ref 46.), runoff due to paleogeography changes up to 30% *greater* than present (Godderis), Cretaceous lithology suggests weatherability was 0-30% *greater* than present (Mills and Kump), and changes in relief suggest that the mid-Cretaceous weatherability was 60-80% that of modern. Given that biologically induced changes are probably modest over the timespan we are considering (see main text), the combination of these weatherability factors is highly unlikely to result in Cretaceous weatherability less than 20% the modern value (e.g. for the most extreme case $0.73 \times 1 \times 1 \times 0.60 = 0.44$). If anything, the 20% cutoff is too generous and the temperature dependence of continental weathering is even weaker than what we report. Furthermore, even if the 20% cutoff is debatable, reducing the cutoff does not dramatically change our results. For instance lowering the cutoff to 10% shifts lower bound on the 90% confidence interval for T_e from 17 K to 16 K – strongly temperature dependent continental weathering is still highly unlikely. A brief summary of this discussion has been added to the 'Model Description' section.

Figures 7 and 8: I think these figures should have cited references in addition to the "code" for each curve (Fig. 7) and/or GCM data point (Fig. 8)? The references seem to be predominantly in the Supplement but presumably should instead be included in the main text, with clear, numbered citations in these figures.

14) Given the large number of studies we cite in these figures, we found it necessary to put these references in the supplement to conform the *Nature Communications'* reference limit. However, we are happy to place the references in the main text if this is the Editor's preference.

Response to Reviewer 3

review of “The global carbon cycle since the mid Cretaceous: implications for climate sensitivity and continental versus seafloor weathering” by Krissansen-Totton and Catling

This contribution aims at showing that the weathering of seafloor basalt weathering is much more sensitive to climate than continental silicate weathering. This has implications on how the carbon cycle operates over the last 100 Myr, and on the Earth system sensitivity to a CO₂ doubling.

Adding one flux in the deep time carbon cycle, the weathering of seafloor basalts (SFW), is an hypothesis that has been tested for the first time in the early nineties (Walker 1990 in P3; François and Walker, 1992). The objective was to decouple the solid Earth degassing and the continental silicate weathering to allow the Sr isotopic composition of seawater to fluctuate. The original models were parametric and, as noted by the authors of the present contribution, they were wrong.

The present paper explore again the role of SFW on the carbon cycle over roughly the last 100 Ma. Similarly to the model developed previously by Le Hir to simulate the CO₂ pumping by SFW during snowball events, they assume that SFW depends on temperature following an Arrhenius type of law. The authors then build a simple model describing the carbon cycle. This model is first run in a direct way. Then it is run following a bayesian inverse method to constrain the values of the 16 main parameters of the model. This inversion is the innovative part of this contribution, leading to interesting results such as the quantification of the required rise in continental weatherability from the Cretaceous to the present day.

However, I have several questions:

1) Everything relies on the observation that SFW was more intense during the mid-Cretaceous climatic optimum than at present day. This is taken from the study of Coogan and Dosso. But this study gives only one point in the past. What about the evolution over the late Cretaceous and the whole Cenozoic ? The authors reconstruct a long term decrease from the high Cretaceous value towards the present day, without supporting data for this scenario. There is a growing evidence that the late Cretaceous was much cooler than the Cenomanian-Turonian episode (see for instance Puceat et al 2007 in Geology). The high mid Cretaceous values might correspond to a short-lived peak, and the inversion method used here would produce drastically different results.

To clarify, the data used from the Coogan and Dosso paper is not a single data point but rather an average of several Cretaceous cores of different ages. These data suggest that the average Cretaceous SFW sink was higher than the average Cenozoic sink (data from the 90 Ma Troodos Ophiolite also supports this conclusion).

However, the reviewer correctly points out that there is considerable uncertainty in the magnitude of the seafloor-weathering sink through time, and the data is sparse enough that we cannot exclude the possibility that we are sampling short-lived peaks. We have our done our best to incorporate these uncertainties into our inversion calculations. The value we are fitting for mid-Cretaceous SFW is a function of spreading rate, which in turn can take on a wide range of values. Consequently, the 1σ range for mid-Cretaceous SFW varies from 2x modern to 5x modern.

More importantly, the constraints we derive for the e-folding temperature of continental silicate weathering, weatherability changes, and climate sensitivity would not be drastically different if the mid Cretaceous SFW values were merely a short-lived peak. In the results section we report the results of a sensitivity analysis whereby seafloor weathering is completely

omitted from our inversion. We find that in this extreme case Cretaceous weatherability is 46^{+16}_{-13} % of modern, as opposed to 42^{+16}_{-12} % when seafloor weathering is included. The differences in T_e and climate sensitivity changes are also minor. Seafloor weathering had only a small influence on results in the submitted manuscript and continues (in the revised version) to have little influence. Instead, weakly T-dependent continental weathering and change in continental weatherability are primarily a result of the model fit to pCO₂, pH, temperature, and saturation proxies. This argument is clearly explained in the revised results section. Of course, our conclusions about the importance of continental vs. seafloor weathering would change if SFW were omitted from the analysis.

2) I would not say that the calculated parameter values are “robust” (this appear at several places in the paper) only because a statistical method has been applied. The results heavily depends on the way equations are written. If the equations are mechanistic, I would agree with the robustness argument. But it is not the case here. There are many parameterizations in the model, some of them new: link between CO₂ and surface T, between surface T and deep T, dependence on pH of the SFW, description of the carbonate deposition in the pelagic environment, continental weathering, the link between CO₂ and DIC... The results are necessarily dependent on the way equation are written.

Reviewer 2 raised the same issue in his/her general comments, and so we refer you to our response number 3 to Reviewer 2 (page 11-12).

3) I feel frankly uncomfortable with the criticism regarding the model of Le Hir 2008. Those critics are wrong and this must be corrected. First, the Le Hir model accounts for the dissolution of the seafloor in acidic environment and in basic environment (it is a sum of terms). Line 630, it is stated that the Le Hir model is in contradiction with the experimental results showing a Ushaped dependence of weathering on pH. But the model of Le Hir uses laboratory kinetic laws fitting the U-shaped curve, so it is fully consistent with experimental results. In Le Hir 08, last line, it is stated that weathering is minimal at pH 8.2. Conversely, the present model does not reproduce the U-shaped curve since it is a function of H to the power gamma. So the question becomes: why do the authors develop a model less efficient than the Le Hir one ? Furthermore, I can't find in Le Hir any discussion about the dominance of basaltic glass dissolution. Where does the discussion from line 625 to line 630 come from ?

The following discussion describes our attempt to reproduce the Le Hir et al. dissolution function. Table 1 in Le Hir et al. (2008, Biogeosciences) provides the experimentally-derived coefficients for calculating seafloor dissolution as a function pH using equation (1) in Le Hir et al. (2008, Biogeosciences). We have written a script that computes dissolution rate as a function of pH based on this equation and these coefficients, and we have plotted the resulting function below. Note that in addition to weighting by the pk_i factors, we have weighted each mineral by its abundance as specified at the end of page 3 (e.g. basaltic glass gets weighted by 9%). The complete dissolution function is written as follows:

$$\begin{aligned}
R_{bas} &= \sum_i f_i k_i \exp\left(\frac{-E_i}{RT_p}\right) a_i^{n_i} \\
&= f_L \times \left(10^{-pk_{L-H}} \exp\left(\frac{-E_{L-H}}{RT_p}\right) a_H^{n_{L-H}} + 10^{-pk_{L-H_2O}} \exp\left(\frac{-E_{L-H_2O}}{RT_p}\right) a_{H_2O}^{n_{L-H_2O}} + 10^{-pk_{L-OH}} \exp\left(\frac{-E_{L-OH}}{RT_p}\right) a_{OH}^{n_{L-OH}} \right) \\
&+ f_D \times \left(10^{-pk_{D-H}} \exp\left(\frac{-E_{D-H}}{RT_p}\right) a_H^{n_{D-H}} \right) \\
&+ f_{BG} \times \left(10^{-pk_{BG-H}} \exp\left(\frac{-E_{BG-H}}{RT_p}\right) a_H^{n_{BG-H}} + 10^{-pk_{BG-H_2O}} \exp\left(\frac{-E_{BG-H_2O}}{RT_p}\right) a_{H_2O}^{n_{BG-H_2O}} + 10^{-pk_{BG-OH}} \exp\left(\frac{-E_{BG-OH}}{RT_p}\right) a_{OH}^{n_{BG-OH}} \right) \\
&+ f_A \times \left(10^{-pk_{A-H}} \exp\left(\frac{-E_{A-H}}{RT_p}\right) a_H^{n_{A-H}} \right) + f_F \times \left(10^{-pk_{F-H}} \exp\left(\frac{-E_{F-H}}{RT_p}\right) a_H^{n_{F-H}} \right)
\end{aligned} \tag{1.1}$$

Here, the subscripts L, D, BG, A, and F represent Labradorite, Diopside, Basaltic glass, Apatite and Forsterite, respectively. Substituting values from Table 1 into this equation yields the following expression for the dissolution of basalt:

$$\begin{aligned}
R_{bas} &= 0.54 \times \left(10^{-8.28} \exp\left(\frac{-65000}{RT_p}\right) a_H^{0.7} + 10^{-11.5} \exp\left(\frac{-68000}{RT_p}\right) + 10^{-9.7} \exp\left(\frac{-50000}{RT_p}\right) a_{OH}^{0.3} \right) \\
&+ 0.32 \times \left(10^{-9.85} \exp\left(\frac{-42000}{RT_p}\right) a_H^{0.14} \right) \\
&+ 0.09 \times \left(10^{-6.70} \exp\left(\frac{-30000}{RT_p}\right) a_H^{0.5} + 10^{-10.35} \exp\left(\frac{-50000}{RT_p}\right) + 10^{-9.45} \exp\left(\frac{-50000}{RT_p}\right) a_{OH}^{0.175} \right) \\
&+ 0.043 \times \left(10^{-pk_{A-H}} \exp\left(\frac{-E_{A-H}}{RT_p}\right) a_H^{n_{A-H}} \right) + 0.013 \times \left(10^{-6.6} \exp\left(\frac{-75000}{RT_p}\right) a_H^{0.5} \right)
\end{aligned} \tag{1.2}$$

We also used $T_p=313$, $R=8.314$ J/K/mol, and the activity of OH is given by, $a_{OH} = 10^{-14}/a_H$. Since $k_H = 10^{-6.7}$ for basaltic glass, it dominates the dissolution function. In the figure below we have also plotted a dissolution $\sim 10^{-0.5 \cdot \text{pH}}$ curve (or equivalently dissolution $\sim a_H^{0.5}$) for comparison, and the basaltic glass curve in isolation. Clearly they are all a very close match, and so this is why we adopted the $a_H^{0.5}$ curve as one endmember case in our analysis. This is also the basis for the discussion from line 625-630. From our analysis, it seems as though the dissolution function provided by Le Hir et al. does not have a minimum at pH=8.2 but rather is monotonically decreasing across the pH range with a slope of ~ 0.5 . However, it is entirely possible that we are misinterpreting the description of the dissolution function in Le Hir et al. 2008, in which case we welcome any corrections. We regret that our original description of the Le Hir model troubled the reviewer and we hope this discrepancy can be resolved.

For the time being, we have simply removed the paragraph about the Le Hir model and instead stated that the pH dependence in our model assumes a range from zero pH dependence to one dominated by basaltic glass.

4) It is also stated on line 104-106 that the Le Hir model ignores the effect of temperature. Once again, this is not true. The Le Hir model includes an Arrhenius law to account for temperature. But, and this is an important point, Le Hir sets the temperature of fluids percolating the oceanic crust to a constant 40°C, while it is set at the deep ocean temperature plus 9K in the present contribution. Alt and Teagle (99) state that the temperature of water-rock interaction fluctuates from 0 to 60°C, depending on the depth of percolation. Because percolation is not modelled in Le Hir (2008), they choose to fix the temperature to a constant value, independently from the deep ocean temperature. Conversely, in the present contribution, the authors assume that the interaction occurs at the deep seawater temperature plus 9K. This 9K value is taken from Coogan and Dosso (2015). Curiously, while all the main parameters of the model are being tested, this 9K value is assumed to be a well-constrained constant. Looking at the Coogan and Dosso paper, this constant can fluctuate from 1 to 20°C. Using the highest value displayed in table 1 for E_a (107 kJ/mol/K) to maximize the effect, I plotted below the factor of increase in SFW when temperature of the deep ocean rises from 3 to 12°C, assuming a correction of 1K, 9K, and 20K. I do not know if this matters or not, but it should be checked.

In the original manuscript we stated that the Le Hir model ignores the effect of temperature as shorthand for the explanation provided by the reviewer i.e. that the model includes a temperature-dependent coefficient, but that temperature is held constant so that changes in temperature have no effect on seafloor weathering. Our original wording was misleading and so we have corrected it to better reflect how temperature is treated in the Le Hir model (the revised manuscript states that temperature was held constant).

In the calculations provided by the reviewer, the flux ratio in the 1K case is only 20% larger than the 20 K case. And even this modest change is for the most extreme case where the activation energy is at the upper end of our range (107 kJ/mol) – for lower activation energies the difference will be even smaller. This variability is small compared to the other sources of error in our analysis, and so for this reason we adopted the point estimate of 9K rather than introduce yet another variable. Indeed, the original text says the following about the +9K temperature modification: “This modification has a very minor effect on the model output

because it is largely the change in temperature, and not its absolute value, that controls variations in the seafloor weathering flux.”

The figures below are forward model outputs for the T=1 K and T=20 K cases, respectively (otherwise identical to Fig. 4 in the main text). It is clear that changing this temperature difference has a very small effect on the model outputs.

Fig: 1 K pore-space temperature difference.

Fig: 20 K pore-space temperature difference

5) line 511-515: the authors may say that they are neglecting the role of organic carbon, but they cannot state that there is no secular change in organic carbon burial fraction over the last 100 My. Check Godderis and François, GRL, 1996; France-Lanord and Derry, Nature, 1997; Galy et al., Nature, 2007. Organic carbon disequilibria are a primary driver of the Cenozoic carbon cycle. Also have a look at Katz et al., 2005, Marine Geology

Organic carbon weathering and burial no doubt plays an important role in the carbon cycle and incorporating an organic carbon subcycle into our model is a future goal. However, to include organic carbon in the present work would necessitate connecting organic weathering to atmospheric oxygen, and consequently would require an oxygen subcycle to track redox sensitive fluxes such as pyrite burial through time. Additionally, we would need to either use the carbon isotope record to infer organic burial rates or make assumptions about biological productivity to infer organic burial fluxes. These changes would introduce many new uncertain parameterizations to a study that is already parameter-heavy.

Instead, we attempt to show that neglecting the role of organic carbon will not affect our conclusions. The Katz et al. paper cited by the reviewer provides reconstructed fractional organic burial (f_{org}) histories based on carbonate and organic carbon isotope records. These globally integrated isotopic signals capture the net organic carbon sink in the Himalayan erosional system quantified by France-Lanord and Derry (1997) and Galy et al. (2007).

The figure above shows the reconstructed f_{org} history from Katz et al. and the best-fit linear trend to this data. From this trend, we conclude that f_{org} has increased by 0.026 over the last 100 Ma, from 0.242 to 0.268. Fractional organic burial, f_{org} , is defined as follows:

$$f_{org} = \frac{F_{org-burial}}{F_{org-burial} + F_{carb-burial}} \quad (1.3)$$

Here, $F_{org-burial}$ is the burial flux of organic carbon (Tmol C/yr) and $F_{carb-burial}$ is the burial flux of carbonates (Tmol C/yr). This can be rearranged to obtain an expression for absolute organic burial:

$$F_{org-burial} = \frac{f_{org} F_{carb-burial}}{(1 - f_{org})} \quad (1.4)$$

Thus, to calculate the inferred change in absolute organic burial we not only need to know the change in f_{org} through time but also the change in total carbonate burial, $F_{carb-burial}$. Since we have neglected organic weathering in our model, we need to add this to our ocean precipitation term, P_{ocean} , to obtain the total carbonate burial implied by our model:

$$F_{carb-burial} = P_{ocean} + F_{weath-organic} \quad (1.5)$$

For now, let's assume organic weathering, $F_{weath-organic} = 7.5$ Tmol C/yr (Holland, 2002, GCA) is constant over the last 100 Ma.

Given the distribution of P_{ocean} curves from our fitted model output (Fig. 5E), it is possible to calculate the inferred change in $F_{org-burial}$ over the last 100 Ma assuming organic weathering has remained constant:

$$\begin{aligned} \Delta F_{org-burial} &= F_{org-burial}(t=0 \text{ Ma}) - F_{org-burial}(t=100 \text{ Ma}) \\ &= \frac{f_{org}(t=0 \text{ Ma})F_{carb-burial}(t=0 \text{ Ma})}{(1-f_{org}(t=0 \text{ Ma}))} - \frac{f_{org}(t=100 \text{ Ma})F_{carb-burial}(t=100 \text{ Ma})}{(1-f_{org}(t=100 \text{ Ma}))} \\ &= \frac{0.268 \times (P_{ocean}(t=0 \text{ Ma}) + 7.5)}{(1-0.268)} - \frac{0.242 \times (P_{ocean}(t=100 \text{ Ma}) + 7.5)}{(1-0.242)} \end{aligned} \quad (1.6)$$

Using this equation and substituting model output values of P_{ocean} we find the median change in the organic burial flux is $\Delta F_{org-burial} = 2.4$ Tmol C/yr, and hence there is a 2.4 Tmol/year imbalance that needs to be compensated for by the other carbon fluxes. However, a sizeable portion of this 2.4 Tmol/year will be balanced by a compensating increase in organic carbon weathering which was assumed constant in the calculation above. Organic carbon weathering is some monotonic function of atmospheric oxygen, and enhanced organic burial will – all else being equal – increase atmospheric pO_2 (e.g. Godderis and Francois 1996). Thus the inclusion of an organic carbon subcycle will introduce a <2.4 Tmol/year imbalance that must be compensated for by changes in the remaining carbon fluxes. A quick inspection of our results (e.g. Fig. 5E, 5F) shows that such an imbalance is smaller than the confidence intervals for all the carbon fluxes we considered. Therefore including an organic carbon subcycle in our model would – at most – shift our distributions by a modest amount.

We have modified the corresponding paragraph in the main text to better explain this argument and reference the Katz et al. paper that the reviewer refers to. We have replaced the line claiming that fractional organic burial hasn't changed with the result that it has only changed by a small amount.

6) The first order controlling factor of continental weathering is runoff, and runoff heavily depends on paleogeographies (Otto-Bliesner, 1995, JGR; Gibbs et al., 1999, P3; Godderis et al., 2014, ESR). How is this dependency included in the present modeling ?

The main text provides the following definition of weatherability:

“The factor ω represents all so-called ‘external’ variables affecting weatherability and encompasses changes in land area due to sea-level variations, changes in lithology, relief, biology, and **paleogeography**. **Rather than attempt to explicitly model these different influences (e.g. ref. 26), we address the inverse problem: how much does ω have to change to fit proxy data?**” (emphasis added)

In other words, the effects of paleogeography are bundled into our weatherability parameter, and so we do not need to explicitly model the runoff-paleogeography relationship. However, the reviewer is correct to highlight the lack of discussion of paleogeography in our interpretation of our weatherability results. Thus in the revised manuscript we have added a few lines discussing paleogeography and we have cited the Gibbs et al. and Godderis et al. papers that the reviewer refers to.

7) line 177: the climate sensitivity evolves through time with the paleogeography (Godderis et al, 2014, ESR). This important effect is neglected here.

Previous inverse modeling of the geological carbon cycle has assumed constant climate sensitivity for simplicity (e.g. Royer, Berner, and Park, 2007, *Nature*). In the absence of detailed modeling of the climate system this is a reasonable first order approach. In the revised manuscript the influence of paleogeography is partly accounted for by modifications to the climate equation, which now includes a secular paleogeography term.

Additionally, we checked to see if allowing for variable climate sensitivity changed any of our key conclusions. We re-ran the inverse model but included two parameters for climate sensitivity, one for before and one for after the deglaciation of Antarctica around ~35 Ma. In this case we are allowing the <35 Ma icehouse world to have a different climate sensitivity to the 35-100 Ma greenhouse world. In this case, the distributions for key variables such as weatherability change and the temperature dependence of continental weathering are very similar to the fixed climate-sensitivity case:

In summary, this is an interesting study, but I think that the above list of questions precludes its publication in Nature Communication. Overall, it remains rather technical and I think it should be submitted to a more specialized journal.

We hope that we have adequately addressed all the questions raised by Reviewer 3. Reviewers 1, 2, and 4 all believe the scope of this paper is appropriate for Nature Communications: Reviewer 1 states “I consider this work significantly more sophisticated than what is typically seen in paleoclimate/paleoceanography papers. I think this paper deserves quick publication and will have a large impact on the field.” Reviewer 2 states “this promises to be a widely read study on a topic of significant interest in the geosciences,” and Reviewer 4 states “This is a subject of great interest and significant prior work, but the current study is perhaps the most conclusive ... These are all important and interesting conclusions, and I recommend publication with modest revision.”

Response to Reviewer 4

Reviewer #4 (Remarks to the Author):

This paper presents a statistically thorough assessment of the factors in the carbonate-silicate cycle that have stabilized the carbon cycle and presumably climate over the last 150 million years. This is a subject of great interest and significant prior work, but the current study is perhaps the most conclusive because of the way Bayesian statistics is applied to assess uncertainties in the parameters and forcing factors. The paper comes to a number of convincing if contrary-to-current-thinking conclusions, including that climate sensitivity of CO₂ consumption during silicate weathering is much lower than previously assumed, that weatherability was lower in the Cretaceous than today (actually this is convention but the treatment here more exhaustive), climate sensitivity was above the IPCC range, and that seafloor weathering, while important (the vanguard thinking) was still dominated by continental weathering during this time interval. These are all important and interesting conclusions, and I recommend publication with modest revision:

1) I'll upload a paper we wrote in 1997 that lays out the concept of weatherability as providing a resolution to the Bernver vs. Raymo arguments about weathering rates and tectonics. In that we calculate a weatherability increase for the Cenozoic that is not unlike that presented here. We also went a bit further to consider imbalances in the organic C cycle, shelf-pelagic carbonate partitioning, etc., implications for Sr isotopes. I'd think the authors might want to cite this on line 42 but also consider comparing our results to theirs later on in the manuscript.

We thank the reviewer for bringing this relevant paper to our attention. The paper is now cited in the introduction (line 42) and we have added a paragraph to the discussion comparing our weatherability results to that of Kump and Arthur. The weatherability factor in our paper, ω , is equivalent to $f_{\text{weath}} * f_a / f_{\text{acarb}}$ in Kump and Arthur (our weatherability factor includes everything except climate-related forcings). We calculated ω through time from Fig. 4 and 14 in Kump and Arthur, and calculated its linear trend (plotted below). Weatherability has increased by ~ 0.3 since 65 Ma according to this figure. If we extrapolate this trend to 100 Ma to match the time span of our model, this implies an increase of 0.46 since the mid Cretaceous, or equivalently, $1+W = 0.53$. Our 1σ confidence interval for $1+W$ extends from 0.3 to 0.58. In short, we find that the weatherability factor calculated in Kump and Arthur is fully consistent with the probability distribution we obtained for the weatherability factor.

Figure: Weatherability coefficient calculated from Kump and Arthur (1997). The linear trend is consistent with the change inferred by the inverse analysis in this study.

2) Line 38 and on: the fact that global weathering rates haven't changed much in the past 12 Ma despite cooling is not inconsistent with the overall carbonate silicate "thermostat" idea: indeed, in the absence of changes in volcanism, silicate weathering rates are predicted to stay constant despite cooling (indicating that the cooling is ultimately being driven by other factors like weatherability).

This sentence has been removed from the revised manuscript.

3) Line 41: I wouldn't lump runoff into "weatherability" since runoff is intimately tied to climate and CO₂; weatherability factors should be external to the climate system (at least in terms of direct relationships).
Runoff has been removed from the list. (We were referring to tectonically driven changes in runoff but we agree the relationship with CO₂ is a potential source of confusion.)

4) Line 58: word missing "Radiometric ... confirms"

Corrected

5) Since the authors are being thorough and precise elsewhere, they need to revise the statement concerning alkalinity (line 63 and on) which incorrectly states that carbonate alkalinity (less water dissociation) must balance the conservative ion charge imbalance: much more important than water dissociation are borate, silicate, phosphate, etc. all the weak acid anions in seawater.

Statement has been corrected. Revised statement says carbonate alkalinity balances conservative ion charge imbalance, neglecting weak acid anions and water dissociation products.

6) The inability to disentangle variations in outgassing from variations in weatherability has a history to it. I think first of the Kasting paper from 1984 (AJS) that showed that the BLAG model prediction of pCO₂ all boiled down to the quotient of outgassing rate to land area (the one "weatherability" factor considered in BLAG). The authors portray this as perhaps more under constrained than it really is: Rowley (1997) has assessed likelihood of spreading rate variations through time, for example.

We assume the Rowley (1997) paper the reviewer is referring to is in fact Rowley (2002) "Rate of plate creation and destruction: 180 Ma to present." This paper presents a reconstruction of crustal production showing relatively little change over the last 180 Ma, thereby implying there has been little change in global outgassing over this time. However, our reading of the literature on outgassing reconstructions does not suggest that the Rowley et al. position is the consensus. For example, Fig. 7 shows a variety of crustal production and spreading rate reconstructions. Many of the more recent constructions such as Van Der Meer et al. (2014) and Verard et al. (2015) imply substantial changes in outgassing over the last 100 Ma. Furthermore, even if we accept the Rowley (2002) position that crustal production hasn't varied over the last 100 Ma, global outgassing has metamorphic and intraplate components that could potentially induce a trend over time. Given these uncertainties, we argue the best approach for this paper is to remain agnostic on the outgassing vs. weatherability issue and allow a broad range of outgassing histories in our model.

7) Can the authors comment explicitly on whether their results argue for transport vs reaction limitation for the present and past worlds, a la Stallard and Edmond 1983?

We hesitate to make any definitive statements on whether our results argue for transport vs. reaction-limited weathering because we are using globally averaged parameterizations to model a diverse range of

environments. However, the weak temperature dependence of continental weathering we derive would seem to argue for transport limitation since experiments yield much stronger temperature dependencies. A sentence has been added to the discussion to make this explicit.

8) We did the basalt exposure through time calculation several years before Mills et al.; see Bluth and Kump (1991; AJS 291:284-308; see also 1994: GCA 58:2341-2359 and for GCM-based weathering calculations last 250 Ma, Gibbs et al. 1999, AJS 299: 611-651; applies to discussion line 399 and on concerning ref 34).

We thank the reviewer for bringing these papers to our attention. Bluth and Kump (1991) are now cited in the discussion on basaltic area through time. The Gibbs et al. paper is also now cited as part of a discussion on paleogeography.

Line 386: no hyphen for ..ly modifiers. Fixed

Line 406: What exactly makes a pCO₂ of 4000 ppm unreasonable? Most of the proxies we use saturate below that level. I don't think we can rule them out.

Wording here (and elsewhere) changed to 'extremely high' rather than unrealistic.

Additional changes

- The title has been changed to meet *Nature Communication's* word-limit and punctuation requirements.
- Since it is a slight tangent, we have moved all discussion of K-feldspar uptake and the comparison with Mills et al. to the supplementary materials to meet Nature Communications word limits.
- The proxy data set has been improved considerably. In the original manuscript we used pH data from Pearson and Palmer (2000), but in the revised manuscript this has been replaced by more recent studies because uncertainties in the Pearson and Palmer data were underestimated (Pagani, M., et al. "A critical evaluation of the boron isotope-pH proxy: The accuracy of ancient ocean pH estimates." *Geochimica et Cosmochimica Acta* 69.4 (2005): 953-961.) The uncertainties in our deep ocean temperatures were reevaluated and increased, and there are slight improvements in our ocean precipitation constraint.
- Proxy data are now binned into 10 Ma intervals and broad error bars are adopted equivalent to proxy ranges within each 10 Ma interval.
- Various minor changes in wording and corrections throughout the manuscript.
- Supplementary figures have been re-numbered and embedded in text.

Reviewers' comments:

Reviewer #1 (Remarks to the Author):

The authors have addressed my concerns. This is an excellent paper that truly advances the field. I recommend swift publication. -Dorian

Reviewer #3 (Remarks to the Author):

Revision of "Constraining climate sensitivity and continental versus seafloor weathering using inverse geological carbon cycle model", by Krissansen-Totton and Catling

The authors have tried to answer most of the questions raised by the first round of review. This represents a non negligible amount of work. However, I think that their contribution still raises some major points.

1) The Bayesian inverse method is a standard statistical method. It allows to define the probability distribution of the parameters of a set of functions, given some constraints on the functions themselves. Consequently, it depends on the mathematical formulations adopted "a priori". The reviewer 2 has the same question. The response of the authors do not convince me:

- the log dependence of surface T on pCO₂ is maybe valid at the global scale. But weathering is made dependent on temperature through a non linear law. This means that global temperature cannot be readily used to estimate global weathering. Furthermore, plenty of teams have produced spatially resolved climatologies for a lot of time slices under various CO₂. I'm afraid to say that equation 5 brings us back in the 90s.
- the law chosen to describe weathering is not convincing: what does represent the direct dependence on CO₂ to the power a ? The impact of soil CO₂ levels ? The only ref for this is Volk (1987). This weathering factor is outdated. Below ground CO₂ levels depend on many unconstrained parameters: CO₂ production by below-ground respiration, temperature-dependent diffusion, tortuosity and porosity, water content. I'm not saying that all this must be included int the model, but this is a major source of uncertainties. We are far from being robust here. Such a factor is also used in GEOCARB prior to the advent of land plants (with a power 0.5) without any ref (I checked in Berner's book, and this parameter was only introduced to avoid too high CO₂ levels in the Paleozoic). For more recent time, Berner was using a Michaelis-Menten rate law. Does this makes any difference ? Finally, why are you not using standard equations from field data, making silicate weathering proportional to continental runoff and to an Arrhenius function of temperature (Dessert et al., 2003; Oliva et al., 2003), instead of speculating on the direct CO₂ factor and its exponent ? Maybe this will produce the same result in terms of the geological evolution of the fluxes.
- I'm really worried about the role of runoff in weathering reactions. The authors elude this question. Line 534 and following: where is runoff in the silicate weathering reactions. Runoff is definitely a key factor: without water, no weathering. Regarding carbonates, temperature plays also a major role, not from a kinetic point of view, but from a thermodynamic point of view (solubility decreases with temperature).
- about weatherability factor w: the authors states that it includes all the external variables affecting weathering (relief, biosphere, paleogeography). But the erosive fluxes are depending on the runoff, the biospheric activity depends on runoff, temperature and CO₂. And the impact of paleogeography occurs through changes in runoff, which depends on climate too. These are not really external parameters. Except of you assume that the dependence of all these factors on climate are lumped into the (pCO₂)^a factor ?
- about carbonate production: carbonate deposition is calculated using quite old parametric laws (Broecker, 1978), which have been calibrated on the present day ocean, and are probably not valid for the geological past. However, I agree that the precise kinetic formulation does not really matter given that the characteristic time of the model is much longer than the mixing time of the ocean.
- in figure 5, silicate weathering and seafloor weathering are both decreasing over the Cenozoic, while total carbonate deposition is increasing. Does this mean that carbonate weathering is rapidly increasing since 100 Ma to close the alkalinity budget ? Why should this be the case ?

2) There is some confusion about the Le Hir contribution. There are two papers in 2008, one in Biogeoscience and the other in Geology. The first one uses a simplified power law (because only acidic conditions were explored), and the second one uses a complete kinetic description for a full

range of pH values, including pH above 8, and displaying a minimal value at pH 8.2. This is a minor point.

3) line 528: I acknowledge the authors for discussing the organic carbon subcycle. They suggest that this cycle plays only a secondary role in the Cenozoic climatic evolution. But they forgot that the carbon isotopic fractionation during photosynthesis has decreased by 8 permil over the Cenozoic (Hayes et al., 1999). This has a pretty big impact on the estimate of the organic carbon sink evolution. I would simply say that this study neglects the potential role of the organic carbon subcycle.

4) there are some odd sentences:

- line 103: the authors state that a complex model does not allow to understand the causal link between processes ? But, by definition, an inverse model does not allow to identify the causal connections.
- line 335 and following: some part of the world are characterized by transport-limited regime. But they are huge surfaces which are still supply limited (all the flat areas in the tropics). Their contributions to the global flux are not negligible because of their size. Check Carretier et al. in *Geomorphology*, 2014. Also, mountain ranges export erosion products in the plains where they are efficiently weathered (check Moquet et al., *Chem Geol*, 2011; Lupker et al *GCA*, 2012). The world is a mix between the supply and transport limited regime.

In conclusion, I think this work is valuable and original. But the authors missed the fundamental point that their results are model dependent. Or they avoid to mention it clearly. They should at least test the silicate weathering model, with the explicit presence of runoff.

Finally, there is something a bit confused in the paper. A large place is devoted to the mathematical formulation of seafloor weathering, and about its potential role. But at the end, the conclusions deal with the sensitivity of continental weathering to temperature changes. The last point of the conclusion is stating that continental weathering is the dominant sink of carbon compared to SFW. But this conclusion might have been reached without any model, by just comparing the present day amplitude of the fluxes. And why should SFW be more important prior to 100 Ma ?

Reviewer #4 (Remarks to the Author):

I feel that the authors have adequately responded to my comments and concerns. Moreover, they did a thorough job of responding to the concerns of the other reviewers as well. I recommend publication as is.

Revisions in the manuscript and supplementary material responding to comments below are highlighted in **green**, whereas previous revisions responding to all four original reviews are highlighted in **yellow**.

Revision of “Constraining climate sensitivity and continental versus seafloor weathering using inverse geological carbon cycle model”, by Krissansen-Totton and Catling

The authors have tried to answer most of the questions raised by the first round of review. This represents a non negligible amount of work. However, I think that their contribution still raises some major points.

1) The Bayesian inverse method is a standard statistical method. It allows to define the probability distribution of the parameters of a set of functions, given some constraints on the functions themselves. Consequently, it depends on the mathematical formulations adopted “a priori”. The reviewer 2 has the same question. The response of the authors do not convince me:

- the log dependence of surface T on pCO₂ is maybe valid at the global scale. But weathering is made dependent on temperature through a non linear law. This means that global temperature cannot be readily used to estimate global weathering. Furthermore, plenty of teams have produced spatially resolved climatologies for a lot of time slices under various CO₂. I'm afraid to say that equation 5 brings us back in the 90s.

To incorporate the effects of spatial variations in temperature on weathering rates would require a climate model that calculates surface temperatures as a function of longitude, latitude, and pCO₂. Adding a model of this complexity would add considerable uncertainty to our analysis (e.g. regional climate sensitivities and paleogeographic boundary conditions). It would also be impossible to perform an inversion requiring millions of forward model calls with an elaborate spatially resolved climate model.

Equation 5 does not represent a return to outdated ideas from the 90s. Numerous recent papers with biogeochemical models adopt similar globally-averaged expressions for continental weathering (e.g. Abbot et al. 2012 ApJ; Mills et al. 2014, PNAS; Mills et al. 2014, G3; Kashiwagi et al. 2008, PPP; Kashiwagi 2016, PPP; Foley 2015, ApJ, and the inverse modeling studies Park and Royer 2011, AJS; Royer et al. 2007, Nature). This is based on the knowledge that, however complicated the regional weathering response to temperature changes may be, we expect an overall monotonic relationship between temperature and weathering due to the kinetics of dissolution and the positive relationship between temperature and runoff.

- the law chosen to describe weathering is not convincing: what does represent the direct dependence on CO₂ to the power a ? The impact of soil CO₂ levels ? The only ref for this is Volk (1987). This weathering factor is

outdated. Below ground CO₂ levels depend on many unconstrained parameters: CO₂ production by below-ground respiration, temperature-dependent diffusion, tortuosity and porosity, water content. I'm not saying that all this must be included in the model, but this is a major source of uncertainties. We are far from being robust here. Such a factor is also used in GEOCARB prior to the advent of land plants (with a power 0.5) without any ref (I checked in Berner's book, and this parameter was only introduced to avoid too high CO₂ levels in the Paleozoic). For more recent time, Berner was using a Michaelis-Menten rate law. Does this make any difference? Finally, why are you not using standard equations from field data, making silicate weathering proportional to continental runoff and to an Arrhenius function of temperature (Dessert et al., 2003; Oliva et al., 2003), instead of speculating on the direct CO₂ factor and its exponent? Maybe this will produce the same result in terms of the geological evolution of the fluxes.

We also cite Schwartzman (2002) in the original manuscript, which provides a derivation of the power law and a discussion of the appropriate exponent for silicate dissolution (the functional relationship between pCO₂ and soil pH is derived, and a range of exponents for appropriate silicates are discussed). Additionally, the supplementary material illustrates the rough equivalence between the simple power law we adopted and more complex models of soil exchange and productivity (Fig. S4 and Volk 1987).

With that said, the reviewer is correct in saying that the functional form for the pCO₂ dependence of silicate weathering – and the overall expression for silicate weathering – is not known precisely. We agree that it makes sense to investigate the sensitivity of our results to different weathering law equations, and so we repeated our inverse analysis using five different continental weathering functions. The results of this sensitivity analysis are referenced in the revised main text and described in full in a new section in supplementary material S7. Despite all the extra work, our key conclusions are unaffected.

In particular, following the reviewer's suggestion, we replaced the power law pCO₂ dependence with a Michaelis-Menton law (case 1). We also adopted the reviewer's second suggestion and omitted any direct pCO₂ dependence and replaced the silicate weathering function with an equation analogous to the field studies cited above (case 2, Arrhenius temperature dependence and linear runoff term). The field studies mentioned by the reviewer are now cited in the new supplementary section. We also included various combinations of these variations such as the original pCO₂ power law with a runoff term, and the Michaelis-Menton law with a runoff term (cases 3 and 4, respectively). Finally, we calculated a case with no pCO₂ dependence and a more generalized runoff dependence (unknown exponent, case 5)

The results are described in full in the revised manuscript, but in summary we find that the functional form of the silicate weathering equation does not dramatically change our key conclusions. The magnitude of the

weatherability increase since 100 Ma is somewhat sensitive to the choice of continental weathering function, and so this caveat is now clearly stated in the discussion and in the conclusions.

- I'm really worried about the role of runoff in weathering reactions. The authors elude this question. Line 534 and following: where is runoff in the silicate weathering reactions. Runoff is definitely a key factor: without water, no weathering. Regarding carbonates, temperature plays also a major role, not from a kinetic point of view, but from a thermodynamic point of view (solubility decreases with temperature).

In the original manuscript, the runoff dependence and the direct temperature dependence in the silicate weathering equation were effectively combined in an overall exponential temperature dependence. This approach is taken in other carbon cycle models e.g. Foley 2015, ApJ; Sleep and Zahnle 2001, GeoSoc. We have modified the model description (section 2) to clarify this.

In the revised manuscript, we perform numerous sensitivity tests where we explicitly separate the exponential kinetic temperature dependence and the linear runoff dependence (see response above). We find that the separating the runoff term does not have a large effect on our results.

The temperature dependence of carbonate weathering was discussed at length in a previous response to a comment by Reviewer 2.

- about weatherability factor w : the authors states that it includes all the external variables affecting weathering (relief, biosphere, paleogeography). But the erosive fluxes are depending on the runoff, the biospheric activity depends on runoff, temperature and CO₂. And the impact of paleogeography occurs through changes in runoff, which depends on climate too. These are not really external parameters. Except of you assume that the dependence of all these factors on climate are lumped into the $(pCO_2)^a$ factor?

Changes in erosion driven by paleogeography or uplift, or changes in weathering due to biological innovations could modulate the temperature dependence of weathering. But this possibility is already represented by introducing a 'catch-all' weatherability factor. This can be seen by taking the derivative of our continental weathering function with respect to surface temperature:

$$\frac{dF_{sil}}{dT_s} = \omega F_{sil}^{mod} \exp\left(\frac{T_s - T_s^{mod}}{T_e}\right) \left[\alpha \left(\frac{pCO_2}{pCO_2^{mod}}\right)^{\alpha-1} \frac{d(pCO_2)}{dT_s} + \frac{1}{T_e} \left(\frac{pCO_2}{pCO_2^{mod}}\right)^\alpha \right] \quad (1.1)$$

Hence an increase in ω since 100 Ma can be interpreted as an increase in the sensitivity of the weathering response to increasing temperature (for example due to uplift enhancing the dependence of runoff on temperature).

This interpretation of the weatherability factor has been discussed previously (e.g. Caves et al. 2016, EPSL).

The $p\text{CO}_2$ and temperature dependent terms in our weathering function do not capture the explicit effects of relief, biology, or paleogeography. Rather, the baseline $p\text{CO}_2$ and temperature dependencies are modulated by a change in weatherability. We have added a few lines to the main text to explain this interpretation: “The weatherability factor, ω , can also be interpreted as the sensitivity of the weathering response to changes in $p\text{CO}_2$ and temperature. For example, an increase in ω implies that an increase in surface temperature will result in a larger change in the continental weathering flux, F_{sil} ”.

- about carbonate production: carbonate deposition is calculated using quite old parametric laws (Broecker, 1978), which have been calibrated on the present day ocean, and are probably not valid for the geological past. However, I agree that the precise kinetic formulation does not really matter given that the characteristic time of the model is much longer than the mixing time of the ocean.

This is a fair point, but as previously stated the precise law for carbonate precipitation will not affect our results. In the revised manuscript we report the results of a slightly more generalized precipitation parameterization where we allow the exponent for seafloor and shelf precipitation to vary independently of one another in the inversion. The results are unchanged by this modification.

- in figure 5, silicate weathering and seafloor weathering are both decreasing over the Cenozoic, while total carbonate deposition is increasing. Does this mean that carbonate weathering is rapidly increasing since 100 Ma to close the alkalinity budget? Why should this be the case? (SEP2) There is some confusion about the Le Hir contribution. There are two papers in 2008, one in Biogeoscience and the other in Geology. The first one uses a simplified power law (because only acidic conditions were explored), and the second one uses a complete kinetic description for a full range of pH values, including pH above 8, and displaying a minimal value at pH 8.2. This is a minor point.

(1) The manuscript states that “carbonate weathering flux at 100 Ma was 18-94% the modern flux (95% credible interval, shown in Fig. 5G)” so yes, the reviewer is correct in stating that our model predicts an increase in the carbonate weathering flux since 100 Ma, although the precise magnitude of this increase is poorly constrained. We have modified the figure caption of Fig 5G to make this more obvious.

We have not highlighted this result because it is largely a consequence of

the functional form chosen for carbonate precipitation in the ocean, which – as the reviewer notes above – doesn't matter too much.

With that said, an increase in carbonate weathering by a factor of ~1.5 is possible. Recall we assumed the same temperature and $p\text{CO}_2$ dependence for both silicate and carbonate weathering. In practice, the temperature and $p\text{CO}_2$ dependence of silicate weathering is likely weaker than that of carbonate weathering. (We account for this in our model by introducing a free parameter for carbonate weatherability that increases since 100 Ma in the inversion, which is equivalent to a weakened temperature and/or $p\text{CO}_2$ dependence for carbonate weathering.) Consequently, the overall change in carbonate weathering flux since 100 Ma is positive – exogenous increases in weatherability dominate over $p\text{CO}_2$ and temperature feedbacks. This may be attributed to sealevel decline exposing more weatherable carbonates, or enhanced physical erosion from uplift. There are no proxies that directly track the carbonate weathering flux, but total carbonate accumulation rates are broadly consistent with a modest increase in the carbonate weathering flux (e.g. Opdyke and Wilkinson, 1988, *Paleoceanography*, Fig. 12).

(2) We are aware of the two Le Hir papers in 2008. The Geology paper does not fully describe the dissolution equation (numerical values for coefficients are not provided in either the main text or the supplementary materials), but the functional form and the mineralogical compositions are identical to that used in the Biogeoscience paper: compare the Geology paper equation A1 and mineralogical assumptions on p. 49 to the Biogeoscience paper equation 1 and mineralogical assumptions on p. 3. The Biogeoscience paper does provide all the necessary information to recreate and plot the dissolution function, which we describe in detail in the previous response to reviewers. Without further information on how the dissolution function in the Geology paper is different to the dissolution function in the Biogeoscience paper, we cannot respond to the points previously raised by the reviewer.

3) line 528: I acknowledge the authors for discussing the organic carbon subcycle. They suggest that this cycle plays only a secondary role in the Cenozoic climatic evolution. But they forgot that the carbon isotopic fractionation during photosynthesis has decreased by 8 permil over the Cenozoic (Hayes et al., 1999). This has a pretty big impact on the estimate of the organic carbon sink evolution. I would simply say that this study neglects the potential role of the organic carbon subcycle.

The 8 permil change in fractionation in photosynthesis is not important for the magnitude of change in fractional organic burial, for (this can be seen in Hayes et al. 1999 Fig. 3d). Crucially, it is for that affects the carbon budget, not the magnitude of the fractionation factor (see Krissansen-Totton et al. 2015, *AJS*, for a review of how carbon isotopes can be related to organic burial fluxes). Whether the organic carbon being buried is sourced from C3 or C4 plants does not impact the rest of the carbon cycle in any direct way.

Our previous comments included a graph showing the observed change in $\delta^{13}C$ since 100 Ma and calculations demonstrating that these variations are too small to drastically change the outputs of our carbon cycle model. The manuscript clearly states our assumptions about organic carbon, “We also do not include organic carbon weathering and burial” and goes on to justify this assumption.

4) there are some odd sentences:

- line 103: the authors state that a complex model does not allow to understand the causal link between processes? But, by definition, an inverse model does not allow to identify the causal connections.

We do not state that a complex model does not allow us to understand causal links between processes, but rather that it is difficult. The cited model (Arvidson et al.) has many complex subcycles and parameterizations that make it hard to understand what is controlling model outputs.

In contrast, we apply both forward and inverse modeling – with a relatively simple forward model – in this study to untangle causal connections. Forward modeling allows us to directly investigate causal connections because we can run the model repeatedly, changing one parameter at a time and observe how model outputs change (effectively a simulated experiment, see section 3.1). Inverse modeling allows us to indirectly investigate causal connections by obtaining posterior distributions for key carbon cycle parameters. These distributions inform our understanding of what is driving observed changes in pCO_2 , temperature, ocean pH etc. (see section 3.2). Additionally, by leaving out observational constraints one-by-one, inverse modeling can determine which constraints are controlling our posterior distributions (section 3.3).

- line 335 and following: some part of the world are characterized by transport-limited regime. But they are huge surfaces which are still supply limited (all the flat areas in the tropics). Their contributions to the global flux are not negligible because of their size. Check Carretier et al. in *Geomorphology*, 2014. Also, mountain ranges export erosion products in the plains where they are efficiently weathered (check Moquet et al., *Chem Geol*, 2011; Lupker et al *GCA*, 2012). The world is a mix between the supply and transport limited regime.

The overall global temperature dependence of silicate weathering is unknown (equivalently, the relative importance of transport vs. kinetically limited regimes is unknown). The papers cited by the reviewer do not provide a definitive answer on this issue. For instance Carretier concludes it is hard to evaluate whether flat areas are important compared to mountains, Moquet concludes there is significant CO_2 consumption from lowland silicate weathering in the Amazon basin, but Lupker finds the Gangetic floodplain

does not constitute a large sink for Ca or Mg, and so is a limited CO₂ sink. The papers we cite in the relevant section of our manuscript argue predominantly for transport-limited weathering, but we acknowledge this presentation is somewhat one-sided. Consequently we have added a caveat to this discussion noting that “the global silicate weathering flux is mixture of transport-limited and kinetically-limited regimes, and so extrapolating from field studies to a global temperature dependence is challenging.”

In conclusion, I think this work is valuable and original. But the authors missed the fundamental point that their results are model dependent. Or they avoid to mention it clearly. They should at least test the silicate weathering model, with the explicit presence of runoff.

We believe that we have parameterized the carbon cycle in a sufficiently general way to be confident in our conclusions. However, we recognize that our results are model-dependent to some degree, and could change if current understanding of carbon cycle processes is overturned. We have made several changes to the revised manuscript to clearly highlight this:

- The abstract prefaces key findings with the caveat “Assuming our forward model is an accurate representation of the carbon cycle...”
- The conclusion section is now prefaced with the caveat “Assuming that proxies are accurate *and that our forward model accurately parameterizes the carbon cycle*, we conclude the following...”

Additionally, the discussion still contains a long section exploring the dependence of our conclusions on the functional forms used to parameterize the carbon cycle (added in response to first round of reviews, and modified to include the new sensitivity analysis).

We have followed the reviewer’s suggestion to investigate the sensitivity of our conclusions to different continental weathering parameterizations. Consequently, we have added a new supplementary material section reporting the results of these sensitivity tests (e.g. with/without runoff and with different direct pCO₂ dependence etc.). We find magnitude of the weatherability change is somewhat sensitive to the functional form adopted, but the conclusions are otherwise unchanged. This caveat is now clearly stated in the discussion section and in the revised conclusions.

Finally, there is something a bit confused in the paper. A large place is devoted to the mathematical formulation of seafloor weathering, and about its potential role. But at the end, the conclusions deal with the sensitivity of continental weathering to temperature changes. The last point of the conclusion is stating that continental weathering is the dominant sink of carbon compared to SFW. But this conclusion might have been reached without any model, by just comparing the present day amplitude of the

fluxes. And why should SFW be more important prior to 100 Ma?

Regarding SFW being more important prior to 100 Ma, the reviewer is correct to point out that this was not explained thoroughly in the original manuscript. We have rewritten a section in the discussion that explains more clearly why SFW may have been important prior to 100 Ma (greater temperature dependence and spreading rate dependence). We have also modified the conclusion to better summarize this.

It was necessary to describe the SFW parameterization in full to establish conclusions about continental weathering. It is not obvious *a priori* that continental weathering dominates seafloor weathering based purely on constraints on SFW magnitude. For instance, Coogan and Gillis 2013, Fig. 5 shows how changes in SFW could have dominated the change in CO₂ consumption since the Mesozoic, depending on outgassing estimates (which are uncertain). Given this paper and others by Coogan, the importance of SFW is a contemporary debate to which our paper provides a worthwhile contribution. Only by applying a self-consistent model of the carbon cycle that incorporates continental and seafloor weathering in a physically plausible way can we make conclusions about the relative strength of these feedbacks. We devote some space to describing SFW because continental weathering parameterizations are well established in the literature.

Reviewer #3 (Remarks to the Author):

Overall, the review improves the paper. The discussion section has been partly rewritten and a discussion about the uncertainties has been added. The supplementary info is now presenting sensitivity tests to various mathematical expressions for continental silicate weathering, as requested in the first round of review.

My main point is that the reference case used in the main text is not the best expression for continental weathering. I strongly suggest that the authors use in the main text the law including a linear function of temperature for runoff and an Arrhenius law for temperature. This is what field data from monolithological catchments show. I really think this might be an important point since most readers never open the SI. And the law presented in the main text is outdated for the following reasons (some arguments come from my first review):

In their pioneering work, Walker et al. mixed laboratory dissolution results to sparse field data. Based on a climate simulation by Manabe and Wheterald (1980, now outdated), Walker adopted the $\exp[(T-T_0)/60]$ factor for the dependence of global runoff on temperature. This was an important step in our understanding of the silicate weathering feedback. More recent modeling works (GEOCARB, various versions) are assuming a linear dependence of runoff on temperature (3.8 % increase in runoff per degree of global warming). A number of 4% has been deduced from the analysis of time series of the major world rivers (Labat et al., 2004). If fitted with an exponential curve, the corresponding e-folding factor becomes 11K, corresponding to an apparent activation energy of 61kJ/mol.

Change in global runoff (normalized to its present day value) as a function of the temperature change. In blue, the original Walker law. In red, a more recent linear fit assuming 4% increase by global warming of 1°C. In black, an exponential fit of the red curve.

Conversely to what is stated on lines 393-394, the runoff-temperature dependence is a function of the paleogeography, even for the Cenozoic. Today, runoff increases by 4% for each degree in global T, but this value falls to 2.5% at 15 Ma. This is the result of a recent 3D climate model (GEOCLIM), explicitly accounting for the paleogeography (Godderis et al., 2014). This dependence is neglected in the present work. It is important to account for the recent developments in the field.

The direct pCO₂ dependence in the weathering law of Walker, used in the present work, comes from laboratory experiments. Lagache experiments have been performed at 2, 6, and 20 bars of CO₂. In modern soils, the maximum pressure can reach 0.01 bars. Walker explicitly mentioned that the Lagache results are probably not valid at natural pressures, but kept them "for purposes of illustration". There is absolutely no reason to keep it in the main text as the reference case.

For all these reasons, I suggest to discuss directly the laws supported by field data in the main text, instead of the outdated Walker formulation which should be moved to the SI.

A few minor points:

line 151-152: the authors compare apparent field activation energies (which includes the runoff-temperature relationship) with laboratory-derived activation energies (which account only for the temperature effect on the dissolution kinetics). These numbers should not be compared.

line 461: the authors acknowledge that their calculated weatherability factor depends "somewhat" on the chosen continental weathering function. I do not know what "somewhat" means, but the authors refer to the SI section 7 for more details. I checked this section, but unfortunately, the effect on weatherability is ultimately "not shown" (line 367). The authors should produce a plot.

line 550: should be "continental silicate weathering" instead of "continental weathering"

Once these last points corrected, I will be happy to support publication of this contribution.

Response to Reviewer #3

In the revised manuscript, blue highlighting shows changes from the latest round of revisions, whereas yellow and green highlighting show the first and second round of revisions, respectively.

Overall, the review improves the paper. The discussion section has been partly rewritten and a discussion about the uncertainties has been added. The supplementary info is now presenting sensitivity tests to various mathematical expressions for continental silicate weathering, as requested in the first round of review.

My main point is that the reference case used in the main text is not the best expression for continental weathering. I strongly suggest that the authors use in the main text the law including a linear function of temperature for runoff and an Arrhenius law for temperature. This is what field data from monolithological catchments show. I really think this might be an important point since most readers never open the SI. And the law presented in the main text is outdated for the following reasons (some arguments come from my first review):

In their pioneering work, Walker et al. mixed laboratory dissolution results to sparse field data. Based on a climate simulation by Manabe and Wheterald (1980, now outdated), Walker adopted the $\exp[(T-T_0)/60]$ factor for the dependence of global runoff on temperature. This was an important step in our understanding of the silicate weathering feedback. More recent modeling works (GEOCARB, various versions) are assuming a linear dependence of runoff on temperature (3.8 % increase in runoff per degree of global warming). A number of 4% has been deduced from the analysis of time series of the major world rivers (Labat et al., 2004). If fitted with an exponential curve, the corresponding e-folding factor becomes 11K, corresponding to an apparent activation energy of 61kJ/mol.

Change in global runoff (normalized to its present day value) as a function of the temperature change. In blue, the original Walker law. In red, a more recent linear fit assuming 4% increase by global warming of 1°C. In black, an exponential fit of the red curve.

Conversely to what is stated on lines 393-394, the runoff-temperature dependence is a function of the paleogeography, even for the Cenozoic. Today, runoff increases by 4% for each degree in global T, but this value falls to 2.5% at 15 Ma. This is the result of a recent 3D climate model (GEOCLIM), explicitly accounting for the paleogeography (Godderis et al., 2014). This dependence is neglected in the present work. It is important to account for the recent developments in the field.

Fig. 13. Runoff change in cm/yr for an increase of 1 °C in the global temperature, calculated by the GEOCLIM model using a three-dimensional calculation of the water budget (solid line). The dotted blue line is the same calculated by the zero-dimensional GEOCARB model, which includes a parametric description of the water cycle. Godd ris et al. / Earth-Science Reviews 128 (2014) 122–138

This is a second order effect for our purposes. Using the numbers cited by the reviewer, and assuming a sizeable 15 K temperature change since the mid-Cretaceous, we have $\text{Runoff}(\text{Cretaceous}) = 1 + 0.025 \times 15 = 1.6$. If the runoff dependence is instead assumed to maintain its modern dependence, then $\text{Runoff}(\text{Cretaceous}) = 1 + 0.04 \times 15 = 1.375$. The constant-runoff dependence case is only $1.6 / 1.375 = 1.16$ times larger than the variable runoff dependence. A 16% contribution from runoff variation from paleogeography changes cannot explain the weatherability doubling we infer, and so the statement in our manuscript is accurate.

The direct pCO₂ dependence in the weathering law of Walker, used in the present work, comes from laboratory experiments. Lagache experiments have been performed at 2, 6, and 20 bars of CO₂. In modern soils, the maximum pressure can reach 0.01 bars. Walker explicitly mentioned that the Lagache results are probably not valid at natural pressures, but kept them "for purposes of illustration". There is absolutely no reason to keep it in the main text as the reference case.

For all these reasons, I suggest to discuss directly the laws supported by field data in the main text, instead of the outdated Walker formulation which should be moved to the SI.

Partitioning the temperature dependence of continental weathering into a linear runoff term and an Arrhenius (exponential) term is useful in some contexts. For example, it is useful for understanding the precise mechanism by which weathering responds to temperature, and for understanding regional heterogeneity in any weathering response. However, in this study we are not interested in elucidating mechanism, but rather constraining the overall temperature dependence of continental weathering from data. For our purposes, *it makes no difference whether we assume an overall exponential or an exponential*runoff function*. This can be seen clearly in the figure below:

This figure shows four weathering functions (black lines) described by the function: $\text{relative weathering} = \exp(\Delta T_s / T_e) \times (1 + \Delta T_s \times 0.04)$. This is the exponential*runoff equation from field data that the reviewer wishes us to adopt. The four lines have effective temperatures of $T_e = 8, 15, 25,$ and 50 K. The four red dashed lines are simple exponentials, $\text{relative weathering} = \exp(\Delta T_s / T_e)$, with effective temperatures chosen to fit the black lines ($T_e = 6.3, 10, 14, 19$ K). We see that for the range of temperature variations we are considering, any exponential*runoff weathering function can be fitted with a single exponential – indeed the reviewer shows this in the figure they provided (although the figure provided by the reviewer does not match their explanation since their fit, $y=0.0329x$ corresponds to an e-folding temperature of $1/0.0329=30$ K, not 11 K as is claimed).

This illustrates that nothing is gained by adding a runoff term to our continental weathering equation. In fact, adding a runoff term unnecessarily complicates the presentation of our results. We could no longer interpret T_e as the overall weathering response to weathering, but would instead need to present the exponential dependence and runoff dependence results separately. We would then need to include additional calculations and/or figures combining the two dependencies into an overall temperature dependence, because in the discussion section we are concerned with the overall temperature dependence and its comparison to previous studies – it is much simpler to use an overall temperature dependence from the start.

With that being said, in the revised manuscript we now better justify the omission of runoff dependence. Specifically, we have now modified the main text to explain that

an exponent*linear_runoff representation of weathering can be accurately approximated by a single exponential. Consequently, we represent weathering as a single exponential such that we can constrain the overall temperature dependence of continental weathering. The revised main text also refers the reader to a new figure (Fig. S1) which illustrates this equivalence.

Regarding the $p\text{CO}_2$ dependence of continental weathering, we maintain that in the absence of accurate quantitative understanding, a power-law $p\text{CO}_2$ dependence is a justifiable approach, and one that has been used by numerous carbon cycle models (e.g. Sleep and Zahnle 2001; Foley 2015; Abbot et al. 2012). The complete omission of any $p\text{CO}_2$ dependence based on field studies is not justifiable because contemporary field studies are all conducted under the same atmospheric $p\text{CO}_2$ level. However, we acknowledge that plausible alternative $p\text{CO}_2$ -dependencies have been proposed – and although we already consider these in the supplementary material – they deserve greater attention in the main text. As a compromise, we have now included the retrieval results for the Michaelis-Menton law $p\text{CO}_2$ dependence (“Case 1”) alongside the reference model case in the main text. For example Fig. 6 now shows posterior distributions for key variables for both the power law case and the Michaelis-Menton case, and the slightly different results are compared in the revised manuscript. Additionally, Table 1 in the main text now describes the posterior distributions of both the nominal model (power law) and the Michaelis-Menton (“Case 1”) alongside one another, such that the full results from both models can be compared. Note also that the range of exponents we have adopted for the Michaelis-Menton law includes the endmember case of no $p\text{CO}_2$ dependence, which is the parameterization the reviewer recommends based on field studies.

Given that we have already demonstrated that changing the weathering function only has a small effect on our results, there is no strong justification for re-running all the sensitivity studies and rewriting the entire paper accordingly. Such an approach would also ignore and sideline the other reviewers of this paper who were satisfied with the first version of the paper with the $p\text{CO}_2$ power law. Reviewer 1 (Dorian Abbot) said, “This is an excellent paper that truly advances the field. I recommend swift publication” and Reviewer 4 (Lee Kump) said “I recommend publication as is.”

A few minor points:

line 151-152: the authors compare apparent field activation energies (which includes the runoff-temperature relationship) with laboratory-derived activation energies (which account only for the temperature effect on the dissolution kinetics). These numbers should not be compared.

We have removed the sentence referring to the laboratory-derived activation energies.

line 461: the authors acknowledge that their calculated weatherability factor depends “somewhat” on the chosen continental weathering function. I do not know what “somewhat” means, but the authors refer to the SI section 7 for more details. I checked this section, but unfortunately, the effect on weatherability is ultimately “not shown” (line 367). The authors should produce a plot.

The line cited by the reviewer refers to the supplementary section on K-Feldspar uptake in the seafloor, not modification of the continental weathering function. The dependence of the weatherability factor on the continental weathering function is discussed at length later in Section S7. Additionally, figures S18, S19, S20, S21, and S22 plot the posterior weatherability factor distributions for our different continental weathering functions, and the 90% confidence factor for the weatherability factors are reported in the respective figure captions. Including an additional figure or table repeating this information would be redundant. We have, however, broken supplementary section S7 into three parts, S7(i), S7(ii), and S7(iii) such that it is easier to find the relevant information referred to in the main text.

line 550: should be "continental silicate weathering" instead of "continental weathering"
Corrected

Once these last points corrected, I will be happy to support publication of this contribution.

Reviewer #3 (Remarks to the Author):

Dear authors,

I thank you for accounting for the last points I raised. The answers to the last questions are convincing. I have no other comments and support publication of this nice contribution.

Best regards.